# Structure prediction of alternative protein conformations

Patrick Bryant [1,2,3] ✉ & Frank Noé [1,4]

Proteins are dynamic molecules whose movements result in different conformations with different functions. Neural networks such as AlphaFold2 can predict the structure of single-chain proteins with conformations most likely to exist in the PDB. However, almost all protein structures with multiple conformations represented in the PDB have been used while training these models. Therefore, it is unclear whether alternative protein conformations can be genuinely predicted using these networks, or if they are simply reproduced from memory. Here, we train a structure prediction network, Cfold, on a conformational split of the PDB to generate alternative conformations. Cfold enables efficient exploration of the conformational landscape of monomeric protein structures. Over 50% of experimentally known nonredundant alternative protein conformations evaluated here are predicted with high accuracy (TM-score > 0.8).

Structure prediction of single-chain proteins is highly accurate for ordered structures with AlphaFold2[1] (AF), RoseTTAFold[2], ESMFold[3] and Omegafold[4]. Generating alternative protein conformations is another important problem which can inform function, but remains unsolved. Several complementary methods to generate alternative conformations with AF-based methods exist. Most methods employ modifications of the multiple sequence alignment (MSA) clustering: (1) using a decreasing amount of sequence clusters[5], (2) a clustering procedure with DBscan[6] to generate diverse clusters[7] and (3) A diffusion model conditioned on the AlphaFold Evoformer embedding[8]. These protocols are evaluated on very few structures (eight, six and five, respectively) whose sequences may be present in the AF training set. Therefore, it is not known if these alternative conformations are already encoded in the AF embeddings.

Other methods to generate alternative conformations are based on molecular dynamics (MD) simulations[9]. The main limitation of this approach is that MD requires vast computational resources to sample the rare-event transitions between long-lived conformations and is, therefore, not scalable to large proteins or complexes[10,11]. Furthermore, most biologically relevant conformational changes in proteins are triggered by physiological changes in the environment, such

as binding or dissociation of a ligand, or phosphorylation of an amino acid, shifting the free energy minimum from one state to another[10,12,13].

To improve upon the previous analyses and provide an answer to whether different conformations can be predicted, we first extracted a set of structures from the PDB[14] that have alternative conformations with substantial changes (a difference in TM-score[15] of at least 0.2 between structures) and are not homologous to the training set of AF. This resulted in a total of only 38 proteins with alternative conformations. We do not think this amount is sufficient to address the multi-conformation prediction problem as the total number of structural clusters (TM-score ≥0.8 within each cluster) is 6696, meaning that only 0.6% of possible structures would be evaluated.

Therefore, we created a dataset suitable for the multi-conformation prediction task by performing a conformational split of the PDB using structural clusters (TM-score ≥0.8). Thereby, we obtain 244 alternative conformations for evaluation which represent all sequences that have nonredundant structures that differ >0.2 in TM-score in the PDB. As AF (and other structure prediction methods) can't be evaluated on this set due to having seen most of these conformations during training, we train a new version of AF on the conformational split which we name Cfold.

[1]Department of Mathematics and Informatics, Freie Universität Berlin, Arnimallee 12, 14195 Berlin, Germany. [2]The Department of Molecular Biosciences, The Wenner-Gren Institute, Stockholm University, Svante Arrhenius väg 20C, 114 18 Stockholm, Sweden. [3]Science for Life Laboratory, 172 21 Solna, Sweden. [4]Microsoft Research AI4Science, Karl-Liebknecht Str. 32, 10178 Berlin, Germany. ✉e-mail: patrick.bryant@scilifelab.se

An important insight is that a network can, likely, not be trained to predict alternative conformations directly based on the coevolutionary information in an MSA. Firstly, there is too little data to develop and assess such a model properly. Although 244 alternative conformations may be enough for evaluation, it is too little for training (thousands are needed[1,16]). Secondly, a structure prediction network should learn to extract coevolutionary patterns that relate to a specific structure[17].

There may be many alternative conformations for a given protein, and using the same coevolutionary representation created from an MSA should not result in different outputs. Therefore, the focus should be on generating different coevolutionary representations that represent different protein conformations, whatever these may be[18]. If an accurate mapping between a coevolutionary representation and a structure has been learned, applying the learned principles to different coevolutionary representations should result in different protein structures that represent a set of present alternative conformations.

## Results

### Predicting alternative protein conformations

Many proteins occupy several different conformations, and each one may be essential for the overall protein function. It is plausible that the information on different protein conformations is embedded in the evolutionary history of proteins and can be extracted through the analysis of multiple sequence alignments (MSAs)[18]. However, one can not know beforehand which part of the full coevolutionary representation in an MSA relates to which protein conformation. Since a structure prediction network is tasked with predicting a single protein structure, obtaining descriptions of multiple conformations through an MSA poses a problem as the network has to choose one.

Here, we train a network to predict only one possible conformation using only sequence information (MSAs). We do this by dividing all single-chain structures in the PDB into structural clusters (Fig. 1a). We then partition the identical sequences that are present in different clusters—the alternative conformations (Methods). Using one of the partitions of conformations, we train a structure prediction network

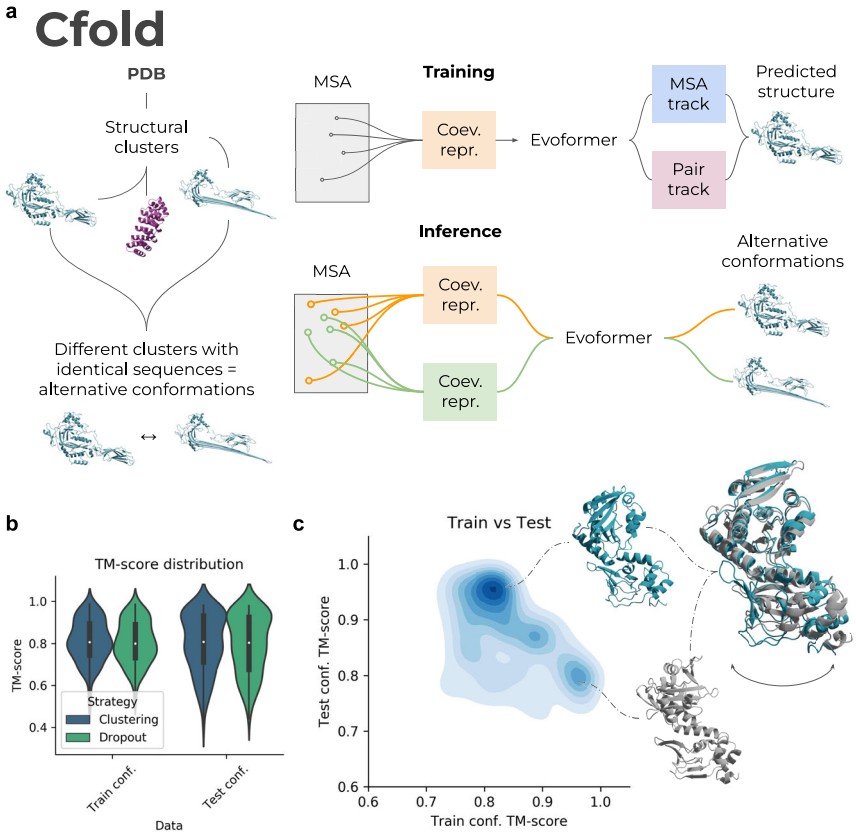

**Fig. 1 | Description of Cfold and test set results. a** Description of Cfold. Using all monomeric protein structures in PDB, we create a conformational split of structural clusters. Different conformations are defined as having >0.2 TM-score differences in identical sequence regions. A structure prediction network is trained on one partition of conformations, and the remaining structural clusters of conformations are saved for evaluation. The network trained to predict structures is similar to the Evoformer of AlphaFold2 (Methods). Two tracks are present, one processing the multiple sequence alignment (MSA) representation and one the amino acid pair representation, the MSA- and pair tracks. At training time, one coevolutionary representation is created to predict one structure. At inference (the MSA clustering strategy is displayed here), the trained Evoformer network is used to predict alternative protein conformations by creating many different coevolutionary representations (orange/green). These are made by sampling and clustering different sequences from the full MSA. **b** TM-score distributions of conformations trained and not trained on (test) for the different strategies (n = 145 and 154 for dropout and MSA clustering, respectively). The MSA clustering results in slightly better results than using dropout. The best TM-score was selected for each method out of approximately 100 samples (Methods). The black boxes encompass data quartiles and the white dots mark the medians for each distribution. The black lines encompass the min/max values. **c** Density plot of the TM-score to conformations in the training set vs. TM-score to unseen conformations using the best strategy (MSA clustering). The higher the density, the darker the colour. Only structures that could be predicted with a TM-score >0.8 for both train and test conformations among the samples taken are displayed (52 structures, n = 5408 sampled predictions). Predicted structures corresponding to lysine acetyltransferase PDB IDs 4AVA/4AVB (blue/grey) are shown.

similar to AlphaFold2[1] and the remaining structural clusters of conformations are saved for evaluation. In contrast to previous attempts to predict alternative conformations[5,7,8], we ensure that our structure prediction network (Cfold) does not see any structures similar to those used for evaluation during training. This is the only way to ensure that predicting different conformations is possible and not an artefact from train and test data overlaps.

Training on structural partitions ensures, in theory, that the network associates one input MSA with one protein conformation. Using the trained network, we then manipulate the statistical representation of the MSA to extract multiple possible conformations described in the coevolution information. We evaluate the trained network on 155 nonredundant structures that can be predicted with the same folds when compared with conformations seen during training (Methods).

To find alternative conformations, we apply two different strategies (Methods):
1. Dropout[19–21]
2. MSA clustering[5]

Dropout has been developed as a regularisation method to train neural networks such that their predictions are more robust. Here we use Dropout at inference time, which leads to different information being excluded at random from each prediction, resulting in different outputs. The rationale for the MSA clustering is similar where sampling different subsets of the MSA (Fig. 1) is thought to generate different coevolutionary representations. This results in different inputs to the network, which, in principle, results in different outputs (alternative conformations). It is also possible that the MSA sampling has a similar effect to Dropout, increasing the stochasticity of the predictions (see below).

MSA clustering proves to be slightly better compared to dropout, resulting in 81 of the alternative conformations being predicted with a TM-score >0.8 (52%, Fig. 1b and Table 1) compared to 76 using dropout. Figure 1c shows a density plot comparing the TM-score of predicted structures towards the training set vs. unseen conformations (test set). The density is higher for unseen conformations at high TM-scores (>0.9), suggesting that the network can predict unseen protein conformations with high accuracy. From all samples, 37% correspond well (TM-score >0.8) to the test conformations, 33% to the train and 30% to neither of the conformations.

## Types of conformational changes

Not all conformational changes are equal. Some are more substantial than others, resulting in e.g. fold switches (alpha helices turned to beta sheets or vice-versa) while others result from a movement of a single domain around a 'hinge'. We categorise conformational changes into three types (Fig. 2):
1. Hinge motion: The structure of all domains stays largely unchanged, but the relative orientation between the domains changes.
2. Rearrangements: The tertiary structure of domains changes, but the secondary structure elements remain largely the same.
3. Fold switches. This is a rare type of conformational change where alpha helices turn to beta sheets or loops or vice-versa.

In total, there are 63 hinge motions, 180 rearrangements and 3 fold switches in the PDB, fulfilling the criteria set for alternative conformations here (TM-score difference >0.2 across identical sequence regions). Among the structures that can be predicted with a TM-score ≥0.6 (155 structures, Methods), there are 40 hinge motions, 114 rearrangements and onefold switch.

**Table. 1 | Number of successful predictions (TM-score >0.8) for conformations in both train and test partitions**

| Method | Train >0.8 | Test >0.8 | Train&Test partitions >0.8 | Total |
|---|---|---|---|---|
| MSA clustering | 81 | 81 | 52 | 154 |
| Dropout | 72 | 76 | 45 | 145 |

The total number represents the proteins that could be predicted out of the 155 evaluated (the remainder failed mainly due to RAM limitations, see Methods).

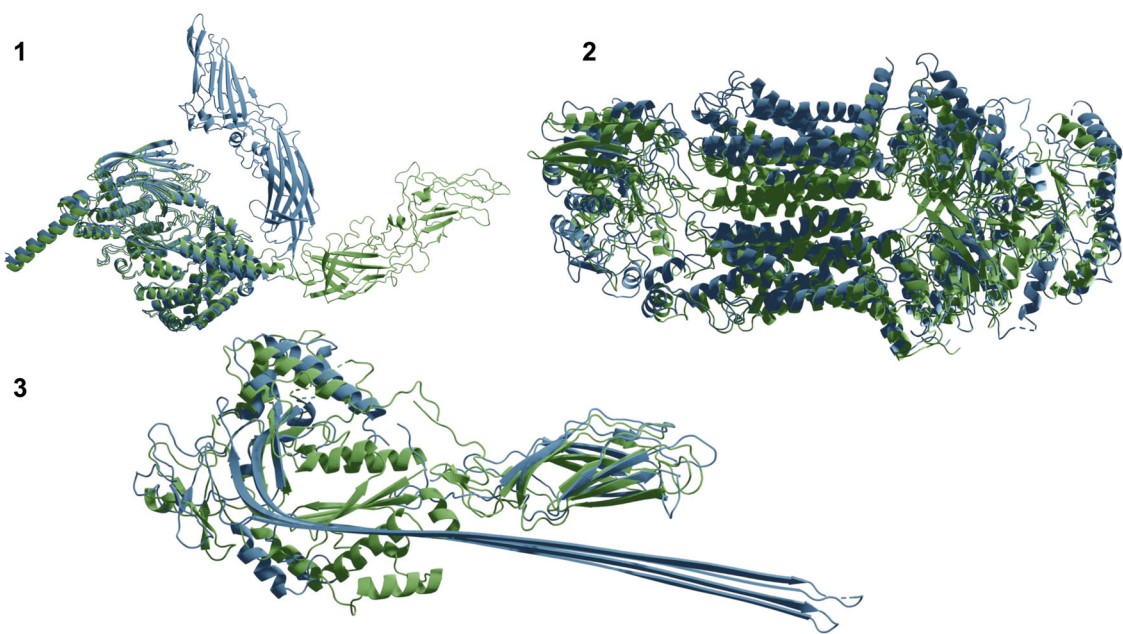

**Fig. 2 | Different types of conformational changes are displayed from native structures.** Hinge motions (1, spike of Bombyx mori cytoplasmic polyhedrosis virus, PDB IDs 7WHM/7WHN), rearrangements (2, surfactant lipid transporter, PDB IDs 7W01/7W02) and fold switches (3, lymphocyte perforin, PDB IDs 3NSJ/7PAG). The most frequent conformational change is the rearrangement (n = 180) followed by the hinge motions (m = 62) and the fold switches are rare (n = 3).

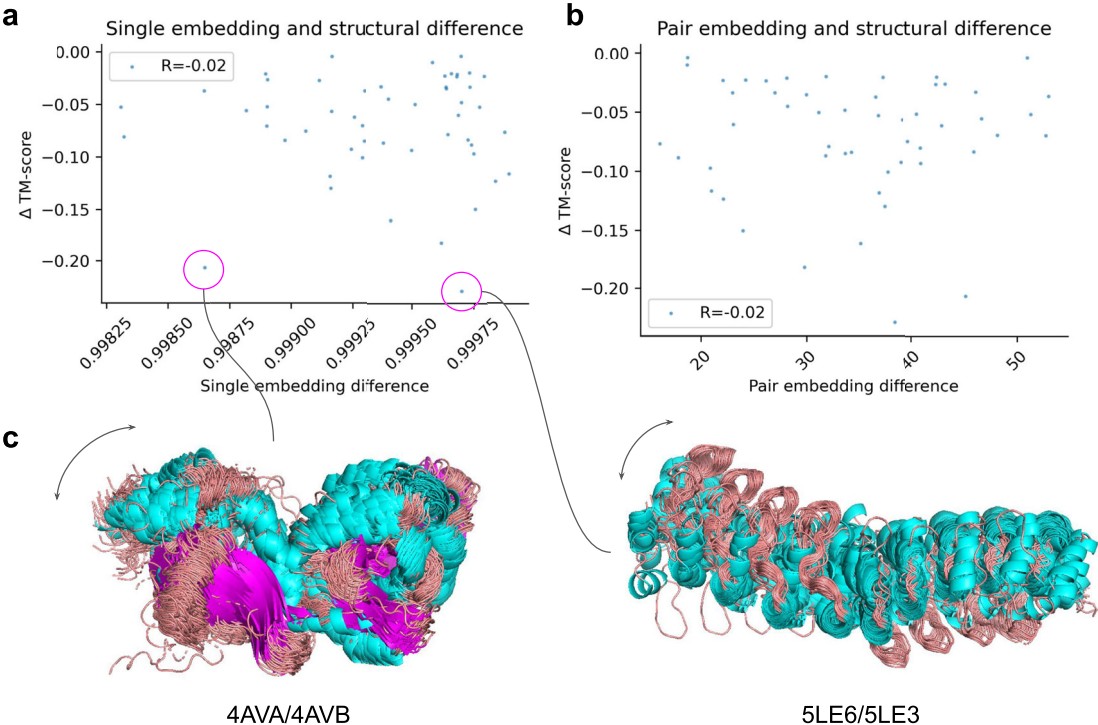

**Fig. 3 | TM-score difference to test conformation vs embedding difference for predictions with the highest TM-scores to the test and train conformations, respectively ($n = 52$). a** Comparison between TM-score and cosine similarity of the single sequence embeddings. **b** Comparison between TM-score and L2 difference of the pair embeddings. **c** Examples of predicted structures coloured by secondary structure (helix=cyan, loop=grey, sheet=magenta) with a big difference in predicted TM-score towards the test conformation, but both small and large cosine similarity. There appears to be no relationship between the variation in cosine similarity and the variation in structural similarity (conformational change).

## The biological relevance of sampled conformations

To connect the sampled conformational changes with biological relevance, we analysed the structures that could be predicted with a TM-score >0.8 for both train and test conformations among the samples taken (52 structures, Fig. 1c). Most structures that display conformational changes interact with a ligand (ligand binding, $n = 42$, Supplementary Table 1), a few are due to introduced mutations ($n = 5$), 3 bind to proteins and for 2 of them, the reason for the conformational variability is unknown, but may be due to ligands which were not resolved in the X-ray data.

The structural changes induced by ligands (mainly binding of substrates to enzymes or transporters) that are important for cellular functions in one way or the other should have states that are accessible through the analysis of coevolutionary information. However, some conformational variation should not be possible to capture through coevolutionary information alone (e.g. elongation factor Tu which gets its conformational variation from binding different ligands, PDB IDs 2C78: https://www.rcsb.org/structure/2C78 and 1HA3: https://www.rcsb.org/structure/1HA3). This suggests that variation in coevolutionary information may be less important than previously suggested[5,7] in exploring possible alternative conformations and that this process is instead stochastic, with different states being constrained by coevolutionary inferred contacts (see below).

## Structural differences are not directly related to differences in internal network representations

Cfold may learn to encode different coevolutionary representations based on the information sampled from the MSA. However, it is also possible that the predictions are stochastic and thereby independent of what coevolutionary representations are provided as input. To analyse these options, we intercept the single embeddings and pair representations before they enter the structure module[1] and compare these.

We select structures that could be predicted with a TM-score>0.8 for both train and test conformations ($n = 52$, Fig. 1c) and rerun the sampling with the MSA clustering strategy to intercept the embeddings. From these, we select the predictions with the highest TM-scores to the test and train conformations and use these as references for the embeddings. We compute the cosine similarity of the single embeddings and the L2 difference between pair representations between the references (Methods).

Figure 3a, b show the embedding differences for the single and pair embeddings vs the TM-score to the test conformation using the selected references (see Fig. 3c for example structures). The Spearman R is only 0.02, suggesting that there is no relationship between the embedding differences and the structural outcome. This indicates that the prediction of conformational states is stochastic and that internal coevolutionary representations are not used to differentiate between states. We note that the coevolutionary differences may not be captured by the cosine similarity or L2 difference used to compare the embeddings and that the network may use the embeddings nonlinearly.

## Structural change and accuracy

The structural change (TM-score between conformations) vs. the prediction accuracy (TM-score) for the test conformations can be seen in Fig. 4a. The accuracy decreases with the structural change, even though there are accurate predictions at changes as large as 0.4 TM-score. Hinge motions are less accurately predicted than rearrangements and fold switches (Fig. 4b). All three types of conformational change can result in both larger and smaller overall structural changes (Fig. 7c), suggesting that the type of change does not determine the accuracy, but the amount of structural change does.

**a**

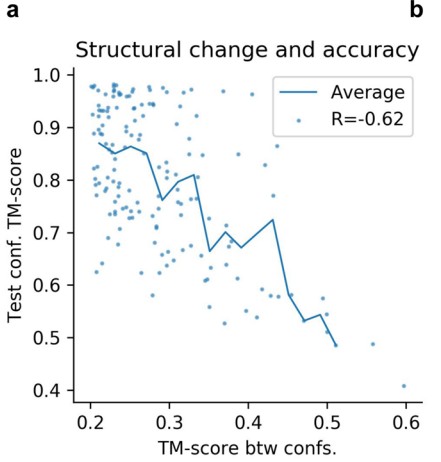

**b**

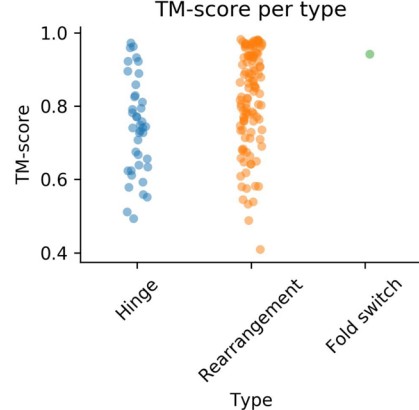

**Fig. 4 | Structural change and conformational type vs accuracy. a** Structural change and accuracy. The x-axis shows the TM-score difference between the native conformations and the y-axis is the TM-score for the prediction and the structure of the test confirmation. The Pearson R is −0.62, suggesting that there is a relationship between the structural change and accuracy and that structures that change more between their conformations are harder to predict. The points represent each structure ($n = 155$) and the line is the running average using a step size of 0.02 TM-score. **b** Conformational type and accuracy. The hinge motions are less accurately predicted than the rearrangements and fold switches. This is likely explained by the findings in (**a**).

## Selecting accurate conformations

The predicted lDDT (plDDT[1]) does not reflect the TM-score towards different conformations (Pearson R = 0.52, Fig. 5a). At the same time, the plDDT score is highly correlated to the overall structural accuracy on the validation set when not considering alternative conformations (Pearson $R > 0.9$, see below Fig. 9c). In the case of alternative conformations, it seems that the plDDT is rather a metric for how the structure may move, describing protein flexibility.

Figure 5b highlights the finding for all samples of structures that could be predicted with a TM-score >0.8 for both train and test conformations ($n = 52$, Fig. 1c). The plDDT scores are high for a range of different conformations and similarities to both train and test conformations vary extensively. There seems to be no clear relationship between the plDDT and known conformations from the PDB suggesting that confidence metrics can't be used to select for certain conformations. This agrees with the findings that this procedure is stochastic (see 'Differences in internal network representations do not correspond to differences in predicted structures').

The regions with lower plDDT tend to be flexible regions that are affected by conformational changes (Fig. 5c). This seems to be the case for predictions for PDB IDs 4AVA/4AVB, 1L5T/2BKA and 7OHG/6S2U. For 6O1X/6O1Z, one of the sampled conformations has low plDDT in the region where the conformations differ and the other has high plDDT values. For 3ZSF/2YLN, both conformations have high plDDT overall and no flexible central region as in 4AVA/4AVB is found. This indicates that the plDDT is to be related to the regions that may change between conformations in some, but not in all cases.

## Discussion

### Structure prediction of alternative conformations with Cfold

The purpose of Cfold is not to replace AlphaFold2 in the prediction of protein structures, but to instead predict more than one conformation of a structure when this is possible. These *conformational ensembles* represent multiple important states of proteins that can inform functional aspects that a single structure can not. In contrast to previous attempts to predict the structure of different conformations[5,7,8], Cfold is evaluated on a comprehensive set ($n = 155$ vs $n = 8$ previously) of structural states distinct from those seen during training. This is crucial to assess if conformational states can be predicted and are not simply reproduced from memory (as is the case when using e.g. AlphaFold2[1]).

Cfold predicts 81/155 unseen structures with high accuracy (TM-score >0.8). Here, 13 samples and three recycles were used per cluster size, resulting in ~100 samples (Methods). However, it has been found that over 20 recycles and thousands of samples are necessary to obtain accurate predictions in challenging cases for multimeric proteins[19]. To see if this may be the case for the single-chain proteins evaluated here as well, we increased the recycles to 10/20 and took 50/100 samples per cluster size (Supplementary information). We see no significant improvement in the scores. Using a higher number of clusters for the predictions is not beneficial either, suggesting that it is best to run the cheaper settings with fewer recycles, fewer samples and a lower number of clusters (Supplementary information).

Proteins with larger structural changes are harder to predict and at changes >0.4 TM-score, the accuracy is poor. This is a limitation of the network, and new methods that do not rely on coevolutionary information may be needed to capture these drastic changes. The type of conformational change does not seem to impact the results, suggesting that if the structural difference between conformations is in the range of 0.2–0.4 TM-score, there is a good chance of obtaining accurate predictions of alternative conformations (57%, 78/138).

### Coevolution and conformational changes

Conformational changes occur due to a variety of reasons, many being direct effects of environmental changes such as interacting with other proteins or substrates. Therefore, it is often not possible to capture them using equilibrium molecular dynamics within one physiological state. Specific knowledge of the exact environmental changes is necessary and we see no realistic possibility of constructing a procedure to enumerate these as anything (including unknown phenomena) may happen in the cell. The reason that conformational states can still be predicted is that they will leave distinctive traces throughout evolution. By extracting these coevolutionary patterns from MSAs it is possible to also extract different conformational states without necessarily knowing what induces them.

However, when we analyse the relationship between internal embedding representations in the network and the structural output, we find no relationship. This suggests that the sampling of alternative conformations is likely stochastic and that the coevolutionary information acts to constrain the structure within a certain outcome. This is also consistent with the finding that larger conformational changes are harder to predict. We also do not find that confidence metrics (plDDT) can be used to select for known conformations as suggested by others[5,22] in contradiction to previous claims[7]. We note that previous findings were deduced by analysing very few structures and that the

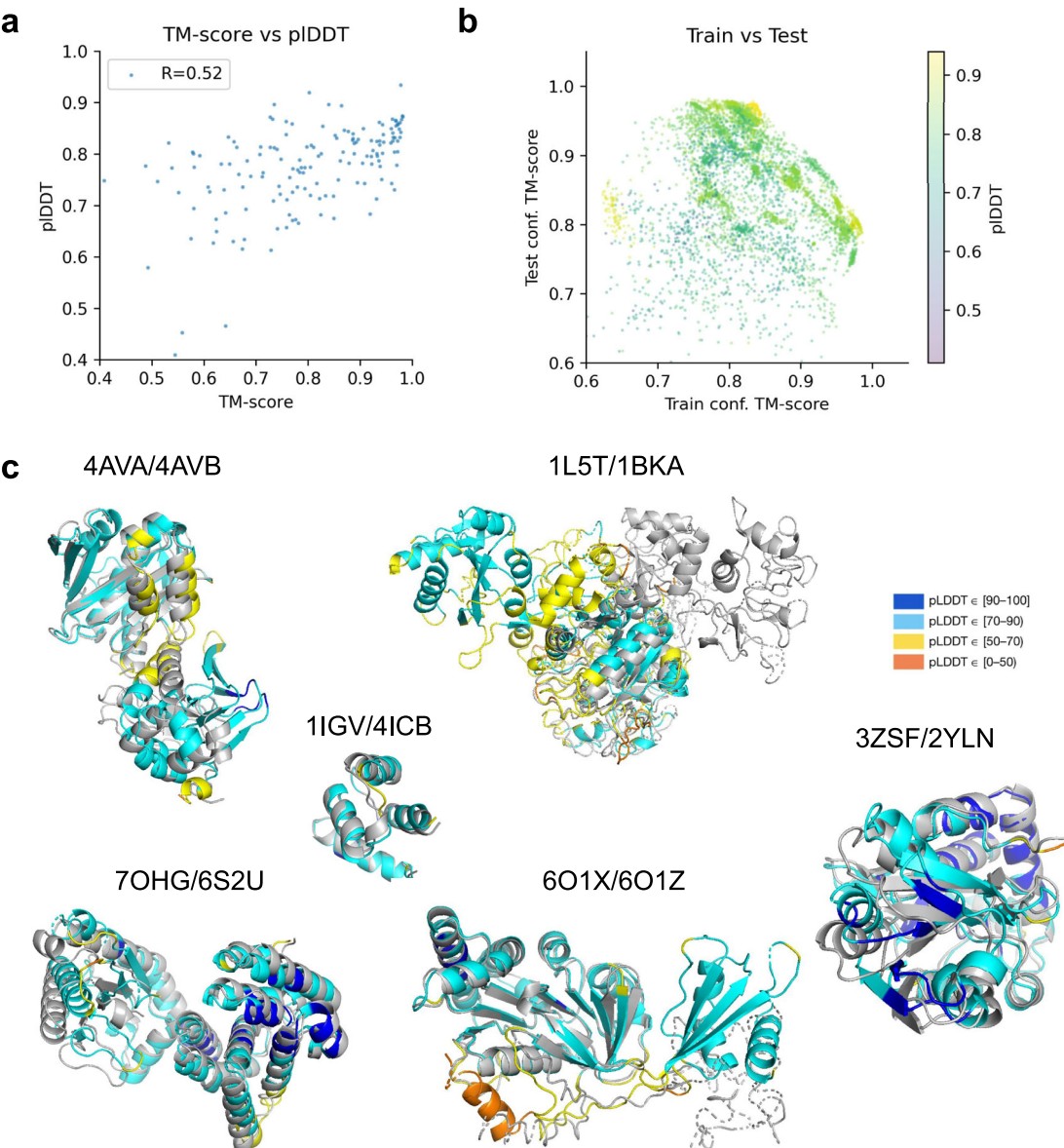

**Fig. 5 | Selecting accurate conformations. a** TM-score vs plDDT. The Pearson correlation between the TM-score and plDDT is low (0.52), suggesting that the quality metric can't be used to distinguish between alternative conformations. **b** TM-score to conformations in the training set vs. TM-score to unseen conformations using the best strategy (MSA clustering) coloured by plDDT. Only structures that could be predicted with a TM-score >0.8 for both train and test conformations among the samples taken are displayed (52 structures, $n = 5408$ sampled predictions). **c** Examples of predicted conformations coloured by plDDT/grey and in structural superposition. Some predictions are less accurate than others, resulting in chain breaks and low plDDT regions (orange).

analysis here should be much more robust as it contains approximately 20 times more data.

## Applications and extensions

As Cfold can predict alternative conformational states with high accuracy in the majority of cases, it is possible to study alternative conformations using available protein sequence data and validate these with experiments for structure determination. We hope that Cfold will aid in extending known alternative conformations as the current pool of 244 conformations out of the 10,116 sequence clusters (2.4%) suggests that there may be many unknown states of proteins. Alternatively, it must be rare for monomeric structures to take on alternative conformational states which poses questions of if dynamical studies are then meaningful for most proteins.

We analysed three structures that display fluctuations in NMR ensembles to see what dynamical aspects Cfold can capture (Supplementary information). We find that Cfold does not capture dynamical aspects of proteins and, therefore, likely not of protein conformations. Cfold can predict distinct conformational states of proteins, but no evidence is found to support the transition between these.

Experimental validation of predictions of alternative conformations would provide the ultimate proof of the utility of Cfold. This would entail predicting the structure of possible conformations of proteins where only one conformation is known, selecting for structural variation and subsequent structural determination by e.g. X-ray crystallography. Such a procedure would require substantial resources and is outside the scope of this study. Importantly, alternative conformations may not be accessible unless these are due to a natural

structural equilibrium. If conformational differences are induced by cellular environments, these may not be possible to observe unless the exact conditions are also determined, making the study of distinct structural states at a scale impossible with current technology.

We expect that Cfold will be useful for a variety of researchers in elucidating various protein mechanisms resulting from conformational changes. To this end, we have made Cfold available through an online application (Google Colab) so that researchers with limited computational knowledge can also apply our method. We aim to extend Cfold to multimeric structures, where more alternative conformations are expected based on varying interaction partners and stoichiometries.

## Methods

### Proteins with alternative conformations not present in the AlphaFold2 training set

AlphaFold2 (AF) was trained on all structures in the PDB[14] with a maximum release date of 30 April 2018. To analyse if AF can predict alternative conformations it is necessary to first obtain structures with conformations not present in this training set. We obtained such a set by:

1. Selecting all monomeric protein structures from the PDB on 2023-07-05 determined by X-RAY diffraction or Electron Microscopy (EM) with a resolution ≤5 Å. We excluded NMR structures as the flexibility in these may give rise to the appearance of alternative conformations when there are none. In total, 69861 structures were obtained. About 51,183 were deposited on or before 30 April 2018 and 18,678 after.

2. We extracted the first protein chain in each PDB file and the corresponding sequences with less than 80% non-regular amino acids and more than 50 residues. 68953/69861 proteins fulfilled these criteria (99%).

3. We clustered the sequences at 30% identity using MMseqs2 (version f5f780acd64482cd59b46eae0a107f763cd17b4d)[23]. This is a more stringent cutoff than the 40% identity used in AlphaFold2[1] to ensure that similar structures are indeed captured within the clusters.

```
mmseqs easy-cluster examples/DB.fasta clusterRes
tmp --min-seq-id 0.3 -c 0.8 --cov-mode 1.
```

4. This resulted in 5452 clusters with more than one entry (10,116 clusters in total). From the sequence clusters

with over one entry, we used TM-align (version f0824499d 8ab4fa84b2e75d253de80ab2c894c56)[15] to perform pairwise structural alignment of all entries within each cluster. We created structural clusters if the pairwise TM-score ≥0.8. 64,208 out of the 68,953 structures (93%) could be clustered into 6696 structural clusters.

5. We now checked for sequence clusters that have different structural clusters (different conformations) before and after April 30 2018. We obtained 900 putative alternative conformations.

6. To see if these are truly alternative conformations and not a result of sequence variations, we clustered the sequences again on 90% identity using MMseqs2:

```
mmseqs easy-cluster examples/DB.fasta clusterRes
tmp --min-seq-id 0.9 -c 0.8 --cov-mode 1.
```

7. We obtained 14,662 clusters out of 64,208 sequences in total. About 7806 of the clusters have more than one member and 64 maps to alternative conformations (some with as many as nine different ones).

8. To see if these are truly alternative conformations and not a result of length differences or low resolution, we performed a manual check. We find that some putative alternative conformations are a result of variations in N/C-terminal loops, mutations or disconnected chains (breaks) and such structures were excluded. The manual check resulted in 38 alternative conformations in total.

### Proteins with alternative conformations in the PDB

To assess all proteins with alternative conformations from the PDB, we continue from step 4 above using the structural clusters at 0.8 TM-score. We skip step 5 and go directly to step 6 (as no date cutoff is necessary here) to see if these are truly alternative conformations and not a result of sequence variations. Figure 6 shows the entire data selection workflow starting from step 1 above.

- After step 6 (clustering at 90% sequence identity), we analyse all sequence clusters at 90% (14,662 clusters) that contain at least two different structural clusters. In total, there are 627 such sequence clusters.

- To reduce the number of necessary manual checks, we do pairwise sequence alignment of all sequences within each structural cluster of the 90% sequence clusters. Using these sequence alignments, we extract the corresponding structural areas and perform another structural comparison with TM-align. Only if the

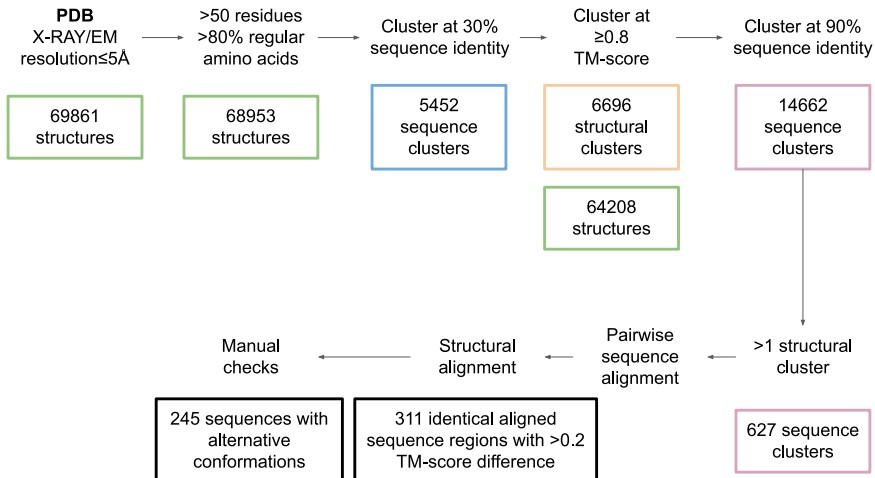

**Fig. 6 | Processing the PDB.** All single-chain structures with at least 50 residues and 20% regular amino acids determined by cryo-EM or X-ray crystallography with a resolution ≤5 Å were selected. These were then clustered at 30% sequence identity and 0.8 TM-score. Within the structural clusters, sequences with 90% sequence identity in different structural clusters were analysed manually to annotate alternative conformations. In total, 245 such sequences were found.

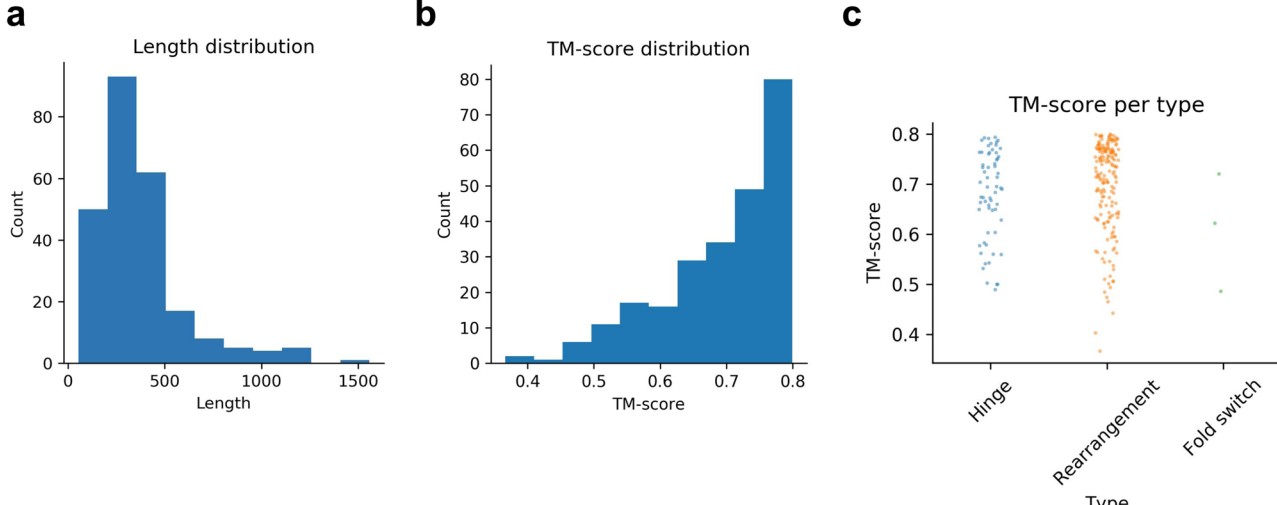

**Fig. 7 | Protein lengths and structural similarities. a** Sequence length distribution. Most sequences are below 500 residues. **b** TM-score distribution between alternative conformations. As most conformations are between 0.7 and 0.8, this suggests that the structural change is mainly 0.2–0.3 TM-score. **c** Strip plot of TM-score between alternative conformations per type (n = 62, n = 180 and n = 3 for hinge motion, rearrangement and fold switch, respectively). The fold switches are fewer with lower TM-scores than the hinge motions and rearrangements.

structures are still different (TM-score <0.8), do we consider them to be putative alternative conformations.

- We obtain 311 such examples that are within the same 90% sequence identity cluster but have aligned sequence regions with corresponding structures that differ >0.2 in TM-score. We select representatives with the longest sequence overlaps and the biggest TM-score difference from each of the structural clusters. We perform a manual check to ensure that the conformations are indeed different and not a result of variable loop regions, low resolution or other experimental differences.

- We obtained 245 sequence clusters at 90% with alternative conformations (TM-score difference >0.2) belonging to 458 different structural clusters. Only one of the conformations of these is selected for training and the remainder is saved for evaluation. We select the clusters at random. In total, there are 6474 structural clusters composed of 59946 sequences for training and 222 composed of 4263 sequences for evaluation with 244 alternative conformations (some structural clusters are present in more than one alternative conformation). We select 315 of the training structural clusters at random for validating the structure prediction performance (5%, 3539 sequences).

Most structures with alternative conformations are below 500 residues and the structural change between conformations is 0.2–0.3 TM-score (see below, Fig. 7). The most represented type of conformational change is rearrangements, and the least fold switches (see below, Fig. 7c). In total, there are 62 hinge motions, 180 rearrangements and threefold switches.

## Training, validation and test partitions
Supplementary Table 3 displays the number of sequences and structural clusters in each partition. Approximately 5% of the sequences and structural clusters were used for validation and testing and 90% for training.

## Scoring
TM-score from TM-align[15] was used for all scoring. We consider conformational changes to have at least a difference of 0.2 in TM-score. This is because smaller changes are very difficult to distinguish. Note that having a TM-score above 0.5 is generally considered as having

similar folds[24] and this analysis is therefore very refined (a TM-score above 0.8 is highly accurate).

Another reason is the evaluation of predicted structures. If a change of only 0.1 TM-score is permitted, the predictions have to have a score of >0.9 to be able to distinguish between the different conformations. This is very unlikely as this means that almost all atoms are coincidental between predicted and native structures. Therefore, we use a threshold of 0.2 TM-score.

## MSA generation
AlphaFold2 generates three different MSAs. This process constitutes the main bottleneck for the predictions as very large databases such as the Big Fantastic Database[25,26] are searched, which is very time-consuming[27]. To simplify this process we instead search only uniclust30_2018_08[28] with HHblits (from HH-suite[29] version 3.1.0):

```
hhblits −E 0.001 −all −oa3m −n 2
```

We note that it is not the number of hits or the number of effective sequences[30,31] in the MSAs that determine the outcome, but rather the coevolution present[18]. This is an elusive concept extracted by the network and it is possible to improve upon the MSA generation at inference by searching larger databases. Indeed, AF has been improved by searching larger sets of metagenomic sequences[32].

## Structure prediction network
**Architecture.** The structure prediction network trained here is almost a complete copy of AlphaFold2[1] for monomeric structure prediction. The main difference in the architecture is that the template track—which processes similar structures—is removed here to focus on the MSA only. Templates can capture alternative conformational states, although this would mean that these states are not predicted but instead copied from these templates.

The configuration for all layers and modules is the same as in AlphaFold2 (Supplementary Table 4).

**Training.** The main difference overall between the network here (Cfold) and other structure prediction networks is how it is trained. We train Cfold on structural clusters of sequences which enables learning a one-to-one mapping of a given MSA and a protein conformation. The

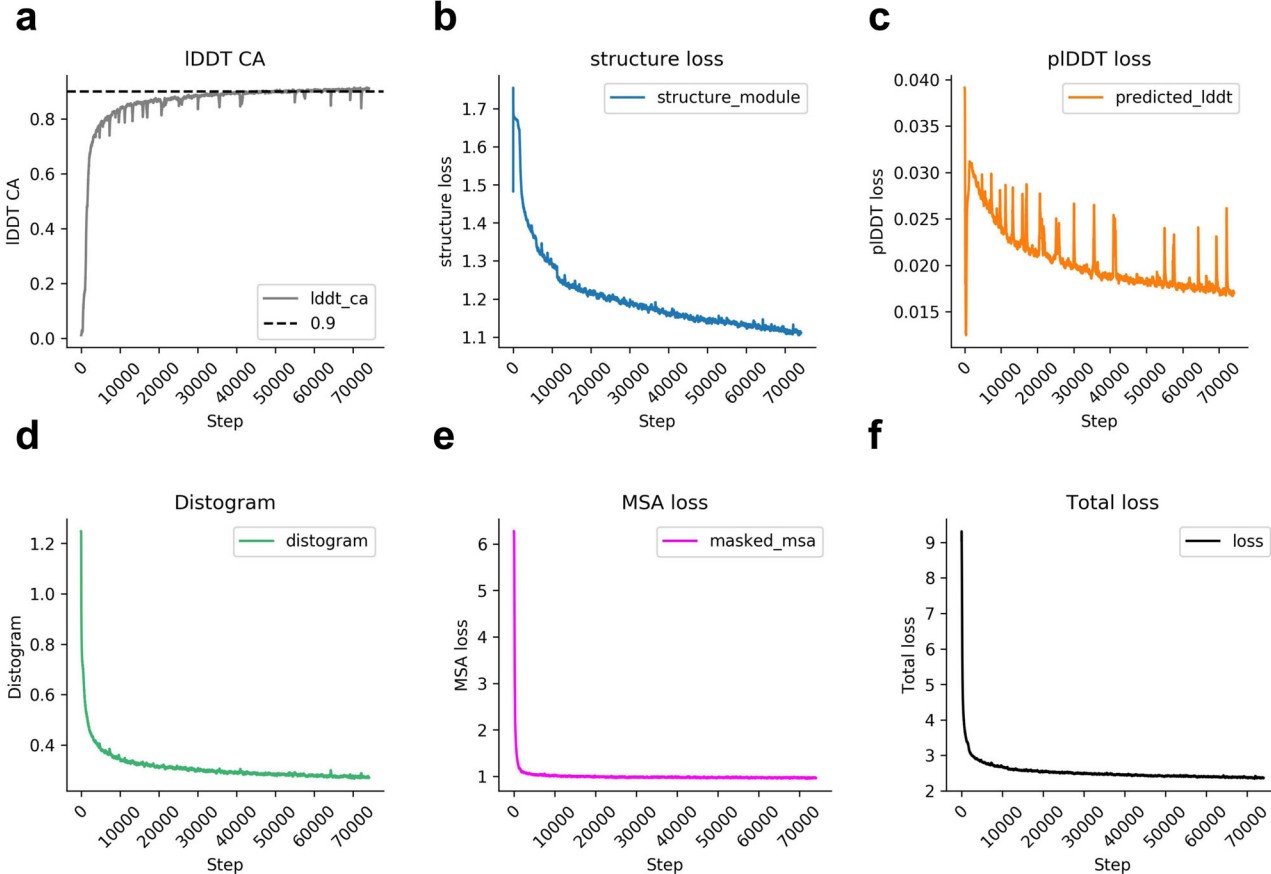

**Fig. 8 | Train losses and metrics vs. training steps.** The losses have been smoothed with an exponential moving average (step size = 100). The lDDT CA increases continuously, although almost all performance is reached in the first 10,000 steps (**a**). First, the MSA (**e**) and distogram losses (**d**) saturate, followed by the structural module loss (**b**) and plDDT loss (**c**). The total loss (**f**) is the sum of distogram, MSA, predicted lDDT (plDDT) and structural module losses.

structural clusters used for training are selected at random, which means that the network has to learn to extract specific coevolutionary information from each MSA that relates to a certain conformation that is learned during training.

This sets the network up to be more focused on one structural description in the MSA, which is important for later evaluating the impact of different MSAs on the outcome. We sample the structural clusters with inverse probability to the cluster size. This allows the network to learn a more refined mapping from each sequence to the exact structure compared to using sequence clusters alone.

The effective batch size is 24 distributed across eight NVIDIA A100 GPUs (three examples per GPU) with a crop size of 256 residues. We apply the same losses as in AlphaFold2:

$$Loss = 0.5 \cdot FAPE + 0.5 \cdot AUX + 0.3 \cdot Distance$$
$$+ 0.2 \cdot MSA + 0.01 \cdot Confidence$$

where FAPE is the frame-aligned point error, AUX *is* a combination of the FAPE and angular losses, Distance is a pairwise distance loss, MS*A* is a loss over predicting masked-out MSA positions and Confidence is the difference between true and predicted lDDT scores. These losses are defined exactly as in AlphaFold2 and we refer to the description there[1].

We use a learning rate of 1e-3 with 1000 steps of linear warmup and clip the gradients with a global norm of 0.1 as in AlphaFold2. The optimiser is Adam[33], applied through the Optax package in JAX, which the entire network is written in (JAX version 0.3.24, https://github.com/deepmind/jax/tree/main). The model is trained until convergence with a total of 74,000 steps (compared to 78,125 in

AlphaFold2). Each step takes approximately 19.7 s, resulting in a total training time of 17 days.

The different losses and how they converge can be seen in Fig. 8. During training, 56,345/56,407 (99.9%) of the training examples could be loaded due to file system issues blocking access to some data. We do not expect that this will impact the performance of the network.

**Structure prediction validation.** To validate the training of Cfold, we use the full-length structures taken from 315 structural clusters (one per cluster sampled randomly) and three recycling iterations. From the 315 sampled structures, 307 can be evaluated (97%) due to structural inconsistencies (e.g. missing CAs) causing errors with TM-align and the lDDT calculations.

Figure 9a shows the TM-score distribution vs. the training step and b the lDDT scores. The best model weights are obtained at 10000 steps and these are the ones used in Cfold. It takes ~3 days to train the model to this point. The relationship between the predicted and true lDDT scores remains stable throughout the training process (Fig. 9c). The median TM-scores at 20,000 and 30,000 steps are similar to those of 10,000, but at ≥10,000 steps, the secondary structure of the beta sheets is inaccurate (Fig. 9d).

**Structural limitations.** To correct bond length violations and overlapping atoms, specific losses are added during the fine-tuning stage in AlphaFold2. In addition, one can obtain slightly more accurate structures by relaxing predictions in the Amber force field[34]. Here, we did not perform fine-tuning as the main objective is to sample globally different conformations. To generate highly accurate structures

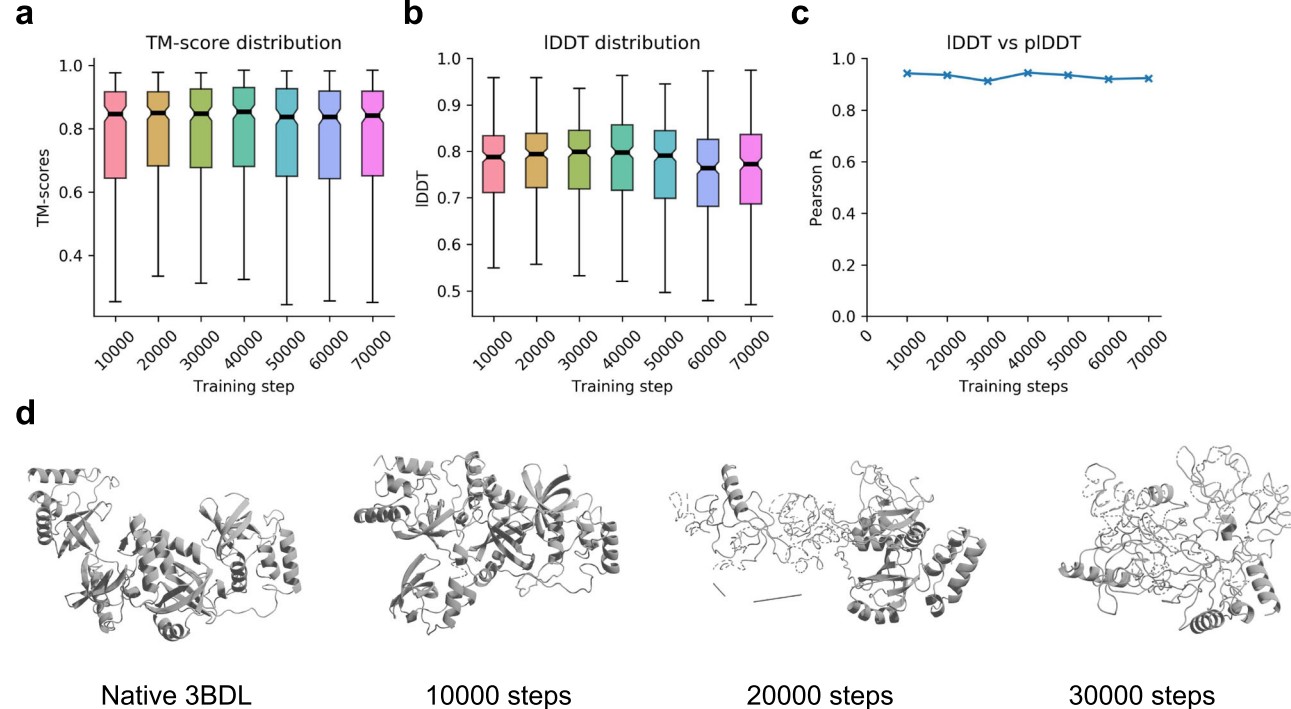

**a** TM-score distribution    **b** lDDT distribution    **c** lDDT vs plDDT

Native 3BDL        10000 steps        20000 steps        30000 steps

**Fig. 9 | Validation. a** TM-score distribution for the validation set vs. training step ($n = 307$). The medians are marked by black lines and the upper and lower quartiles are in colour. **b** lDDT CA distributions vs. training step ($n = 307$). **c** Pearson correlations for lDDT vs. plDDT for the alpha carbons (CA, $n = 307$). The correlation coefficients ($R$) remain close to 0.9 throughout the training. The medians are marked by black lines and the upper and lower quartiles are in colour. **d** Predicted structures for PDB ID 3BDL from the validation set at different training steps. The TM-scores are 0.59, 0.53 and 0.55 for 10,000, 20,000 and 30,000 steps, respectively. Visually, one can see that the network starts to make worse predictions >10,000 training steps, although this is not apparent from the metrics in (**a**, **b**, **c**). This suggests that the network starts to overfit certain structures at an early stage.

locally, the same Amber force field can be applied to obtain consistent side chain angles for all amino acids. We did not relax the structures here as this will likely not impact the overall performance.

### Conformational generation

We select structures whose conformations are in the train set from the test clusters that Cfold can predict with a TM-score of at least 0.6 using one prediction (0.5 is considered to be of a similar fold[24]), resulting in 155 out of 234 possible structures (66%). One reason that not all structures are predicted with the same fold as would be expected from AlphaFold2 is that Cfold is trained with fewer data (excluding e.g. monomers extracted from multimers) for a smaller number of steps. Other confounding factors may be that these structures do indeed have alternative conformations, and the training states may not be favourable.

We take the minimum TM-scores from the TM-align structural superpositions here. To analyse if Cfold can predict the structure of alternative conformations of these, we apply two different strategies:
1. Dropout[19–21]
2. MSA clustering[5]

For the dropout, we simply activate dropout everywhere except for in the structural module following AFsample[19] and make 100 predictions in total. For the MSA clustering, we set the number of clusters used in the sampling for the predictions to vary between [16, 32, 64, 128, 256, 512, 1024, 5120][5]. We take 13 samples per clustering threshold, resulting in 104 predictions per target in total (compared to the 50 used previously). In total, 145 and 154 out of 155 structures could be predicted using strategies 1 and 2, respectively. The number of resulting structures sampled is 14177 and 16007 for strategies 1 and 2, respectively. The failures are due to memory issues (NVIDIA A100 GPUs with 40 GB RAM were used).

We evaluate the predictions with the TM-score from TM-align, this time taking the maximum score to account for possible size differences between the native structures representing the different conformations (extracted matching regions) and the full genetic sequences used for prediction.

### Embedding comparison

Out of 10,816, 10774 structures were sampled successfully (99.6%, the failures were due to memory issues for large cluster sizes using NVIDIA A100 GPUs with 40 GB RAM). For the single sequence embedding, we calculate the cosine similarity using the models which best correspond to train/test conformations:

$$Cosine\ similarity = \frac{A \cdot B}{|A| \times |B|} \cdot \frac{1}{L} \quad (1)$$

where A and B are single sequence embeddings and |A| and |B| represent the two-norms and L is the protein length.

For the pair embeddings, we instead compute the pairwise differences overall:

$$Pairwise\ difference = \frac{1}{n} \sum_n \sqrt{(C - D)^2} \quad (2)$$

where C and D are the pair embeddings and n is the number of elements in the matrices.

### Reporting summary

Further information on research design is available in the Nature Portfolio Reporting Summary linked to this article.

## Data availability

The data that support this study are available from the corresponding authors upon request. The data used to train the network and PDB files of the predictions used to generate the figures are available at: https://zenodo.org/records/10837082. The source data underlying the Figures are also provided as a Source Data file. Source data are provided with this paper.

## Code availability

Cfold is available for local installation here: https://github.com/patrickbryant1/Cfold and as a Google Colab notebook here: https://colab.research.google.com/github/patrickbryant1/Cfold/blob/master/Cfold.ipynb.

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

## Acknowledgements

This study was supported by the European Commission (ERC CoG 772230 "ScaleCell", F.N.), MATH+ excellence cluster (AA1-6, AA1-10, F.N.), Deutsche Forschungsgemeinschaft. (SFB 1114 projects AO4 and C03, RTG 2433 Project Q04, SFB/TRR 186 Project A12, F.N.), the BMBF (Berlin Institute for Learning and Data, BIFOLD, F.N.) and SciLifeLab & Wallenberg Data Driven Life Science Programme (grant: KAW 2020.0239, P.B.). Computational resources were obtained from ZIH (SCADS) at TU Dresden with project id p_scads_protein_na (P.B. and F.N.) and from LiU with project IDs Berzelius-2023-267 and Berzelius-2024-78 (P.B.). Blender 3D and Pymol were used to visualise the protein structures. We also thank DeepMind for making the AlphaFold2 code available and Atharva Kelkar for reading the manuscript.

## Author contributions

P.B. designed and performed the studies, prepared the figures and wrote the initial draft of the manuscript. All authors contributed to reading and improving the manuscript draft. F.N. obtained funding. P.B. and F.N. obtained computational resources.

## Funding

## Competing interests

The authors declare no competing interests.
