## [Peer Review File · Nature Communications]

Structure prediction of alternative protein conformationsREVIEWER COMMENTS

Reviewer #1 (Remarks to the Author):

This paper focuses on the prediction of alternate protein conformations via machine learning. A number of previous efforts have addressed the challenge by modifying input to the AlphaFold2 pipeline, with sampling of multiple sequence alignments being a promising strategy. Here, a new AlphaFold2-like model is trained and applied in a similar manner by either sampling sequence alignments or applying dropout functions to obtain conformationally varying structures for a given sequence.

The main motivation for retraining a new network is to obtain a larger number of independent test cases on which AF2 has not been trained before and, as in previous studies based on AF2, there is success in obtaining structural models of alternative states. The analysis is interesting, but at the end it seems to only confirm that sampling multiple sequence alignment space, e.g. via clustering, is indeed a useful strategy to sample conformational states via AF2.

The manuscript is difficult to understand at many places and it was often not quite clear to me what exactly was done and how the results should be interpreted. Some specific questions I have are given below:

- 1) For which systems were alternate conformations generated and evaluated and what was present in the training set? Were alternate states predicted for structures present in the training set or for entirely different sequences with none of the alternate conformations seen during training. And were any structures similar to the predicted alternate states present in the training set (irrespective of sequence similarity)?
- 2) The overall prediction accuracy of Cfold seems unclear. On one hand Figure 8 suggests 'good' performance (although unclear compared to AF2), but then it seems that only 66% structures were predicted with $TM > 0.6$ for the alternate state predictions. What does that mean?
- 3) It seems that what is reported are the best conformations from a large set of samples with no clear mechanism for selecting such conformations without knowing the answer. If so, that would be a significant limitation to practical applications.

Reviewer #2 (Remarks to the Author):

The work by Bryant and Noe is a timely and pertinent experiment that is exactly set up to address many current open questions about deep learning models and the prediction of multiple conformations.

However, in my opinion the work is currently missing the big picture for interpreting their results. To be fit for publication in Nature Comms, the authors need to dig deeper into their results in the following major ways, to provide the field with more insight in the prediction of multiple conformations.

The work is also in general lacking connections to biological function of the discussed proteins, and this needs to be significantly improved in my view to be fit for publication. For instance, the example of perforin is used frequently throughout the paper as an example of predicting multiple conformations. The structure 7PAG is one subunit of a symmetric multimer (see <https://www.science.org/doi/10.1126/sciadv.abk3147>, Fig 1C). If 7PAG was in the test set, then this means that learning from [primarily] monomers enabled Cfold (and AF2) to predict structure in both a monomeric and multimeric context. None of the above is described in the paper currently.

1 It has been posited by others [see, for instance, introduction of <https://www.nature.com/articles/s41586-023-06832-9>] that dropout in AF2 is able to sample multiple conformations when evolutionary couplings for both structures are present at the level of an entire MSA, and dropout or subsampling an MSA obstructs one set of contacts, allowing for the other to dominate driving the prediction. As long ago as 2012 in Hopf et al. Cell, it was shown that evolutionary couplings were present for multiple sets of contacts of ion channels.

The authors have the potential to evaluate if this is true more generally using their network. For a

given protein where both conformations are correctly predicted, can the authors discern if embeddings from the evoformer that lead to either state are significantly different before they are fed into the structure model? If so, this indicates that the evoformer has already learned differentiating representations for both. If not, it would indicate that prediction of both states is stochastic.

2) Dropout is statistically principled to represent underlying uncertainty within a model, see, for instance, <https://arxiv.org/abs/1506.02142>. Therefore, I was intrigued to see sampling comparing dropout and MSA clustering, which is in effect taking fewer random sequences from across the MSA. These two sampling methods should have fundamentally different theoretical bases, with dropout representing uncertainty across the entire model and MSA clustering adding noise at the sequence level.

With this in mind – I want to know more about the proteins that showed a difference between MSA clustering and dropout, and if increased sampling there recovers the failed predictions from dropout. If not, can the authors identify why these worked for MSA clustering and not dropout?

Sampling could also be used to interpret the AF2/Cfold framework. For instance, can the authors use dropout just in some parts of the model to pinpoint more precisely how the AF2 framework is leading to predicting multiple conformations? (i.e., is it in the pair representations? Is it in the first vs 10th evoformer block?) This is probably beyond the scope of this paper, but just floating the idea.

3) The authors could do more in their interpretation of the process of sampling. My guess is that this sampling is far from sampling a Boltzmann distribution from each of the targets, but just how far? Specifically it would be interesting to report what proportion of samples correspond to the trained, vs alternate state, vs. other states. It should be quite possible to compare these to literature experimental values for at least a handful. This would be an opportunity to dig more into biological insight in AF2 + multiple conformation prediction.

4) Fig 1c presents an interesting result – if I understand right, it seems that some proteins resulted in the “seen” conformation being sampled more, whereas others result in the “unseen” conformation being sampled more. Can the authors interpret any attributes of what causes proteins to fall into either of these extremes? This would be another potential place for interesting insight.

5) I recognize that Fig. 4 is making the case that pLDDT is not predictive of model accuracy, but the authors need to discuss / show statistics on pLDDT vs tmscore for within the sampling for each member of the test set. If pLDDT is not correlated with TMscore, can the authors provide any insights/takeaways on how to select models for prospective tests?

6) Depending on how resource intensive training is, the authors could use their setup to answer outstanding questions on how much “regular” AF2 has memorized multiple conformations by comparing the current Cfold to a second re-training of Cfold with all the structure clusters in the train set. This is not a hard ask from me, but if it were easy to re-run it would elevate the impact of the paper.

Minor:

Throughout the paper, proteins are referred to by their PDB codes. They need to be referred to by their actual names.

I got hung up on when TM-score is referring to TM-score to the ground truth and TM-score between multiple conformations. Would recommend finding a way to better differentiate phrasing.

“the structural change between conformations is 0.2-0.3 TM-score (Figure 6)” I don’t understand how 0.2-0.3 TMscore is represented in Fig 6. To me it looks like most TM-scores are around 0.8? Please clarify the writing.

The writing is overly simplistic at times:

“Conformational changes are direct effects of environmental changes” -> this is simply not true, discounted by proteins that in vitro interconvert between distinct states. I get the sentiment but it can be made more accurately.

“Currently, protein structure is viewed as a single snapshot in time while in reality, it should be viewed rather like a video. This video is the continuous collection of states that make up the protein and its functions.” Again while I agree with the sentiment, this is overly simplistic -- “protein structure is viewed as a single snapshot in time” – 50 years of spectroscopists beg to differ!

Note in methods how long training took and on what sort of hardware.

Dear Editors and Reviewers,

We thank you for your insightful comments and suggestions and here provide a point-by-point response to all raised issues and suggestions in bold. We have updated the manuscript accordingly and marked all changes in red throughout the text.

REVIEWER COMMENTS

Reviewer #1 (Remarks to the Author):

This paper focuses on the prediction of alternate protein conformations via machine learning. A number of previous efforts have addressed the challenge by modifying input to the AlphaFold2 pipeline, with sampling of multiple sequence alignments being a promising strategy. Here, a new AlphaFold2-like model is trained and applied in a similar manner by either sampling sequence alignments or applying dropout functions to obtain conformationally varying structures for a given sequence.

The main motivation for retraining a new network is to obtain a larger number of independent test cases on which AF2 has not been trained before and, as in previous studies based on AF2, there is success in obtaining structural models of alternative states. The analysis is interesting, but at the end it seems to only confirm that sampling multiple sequence alignment space, e.g. via clustering, is indeed a useful strategy to sample conformational states via AF2.

The manuscript is difficult to understand at many places and it was often not quite clear to me what exactly was done and how the results should be interpreted. Some specific questions I have are given below:

1) For which systems were alternate conformations generated and evaluated and what was present in the training set? Were alternate states predicted for structures present in the training set or for entirely different sequences with none of the alternate conformations seen during training. And were any structures similar to the predicted alternate states present in the training set (irrespective of sequence similarity)?

The idea with Cfold is to provide more support for that structure prediction networks (AlphaFold2) can be used to predict alternative conformations. Previously, although some indications have been provided for this, such an evaluation has not been possible as almost all conformations have been seen when training AlphaFold2. Therefore, any claims made by using AlphaFold2 are not confirmed but rather given support for the first time here. Cfold includes almost 20x more data than previous analyses and contest findings of how structure prediction and selection of alternative conformations work.

2) The overall prediction accuracy of Cfold seems unclear. On one hand Figure 8 suggests 'good' performance (although unclear compared to AF2), but then it seems that only 66% structures were predicted with $TM > 0.6$ for the alternate state predictions. What does that mean?

We can't directly compare to AlphaFold2. The performance should be identical to AlphaFold2 if this network had been trained to the same degree (as the network architectures are identical). AlphaFold2 has been trained on more data than Cfold though - including e.g. monomers extracted from multimers and all possible alternative conformations.

As less data was available and a fewer number of steps were taken for training Cfold, the resulting confidence will not be as high on all protein structures. Other confounding factors may be that these structures do indeed have alternative conformations. Still, we excluded structures whose conformations present in the training set we could not capture with a certain degree of confidence (TM-score 0.6).

We have added a paragraph about this in Methods to acknowledge this limitation:

“One reason that not all structures are predicted with the same fold as would be expected from AlphaFold2 is that Cfold is trained with less data (excluding e.g. monomers extracted from multimers) for a fewer number of steps. Other confounding factors may be that these structures do indeed have alternative conformations and the training states may not be favourable.

”

3) It seems that what is reported are the best conformations from a large set of samples with no clear mechanism for selecting such conformations without knowing the answer. If so, that would be a significant limitation to practical applications.

Indeed, we do not provide any clear way of selecting these conformations. However, we do not only report the best conformations. As can be seen in Figure 1c, we display a KDE plot of all sampled conformations. Here it can be seen that the test conformational state is favourable as the density is much higher there. For Figure 1b, the reviewer has a point though and we acknowledge that this is a limitation. As it seems to be likely to capture informative states (Figure 1c) we still believe the method to be useful for exploring potential changes within protein structures.

We have tried to make this clear by visualising the TM-score distributions for train and test conformations in Figure 1c (density plot). We also discuss this limitation under “Selecting accurate conformations”. In addition, we note that AF can't be used for selecting conformations as has been found recently here:

<https://www.biorxiv.org/content/10.1101/2024.01.05.574434v1>. This critique underlines the problems with using AF directly on conformations it has seen during

training - the reason for developing Cfold. Previous attempts to predict alternative conformations seem to have had a substantial amount of luck as this only works in a single case with very little structural difference between states (https://www.nature.com/articles/s41586-023-06832-9?utm_campaign=related_content&utm_source=HEALTH&utm_medium=communities) and the reason for why is not clear. This is also a reason for the investigative nature of Cfold which shows that we cannot reliably select different conformations only stochastically sample them with a certain probability.

Reviewer #2 (Remarks to the Author):

The work by Bryant and Noe is a timely and pertinent experiment that is exactly set up to address many current open questions about deep learning models and the prediction of multiple conformations.

However, in my opinion the work is currently missing the big picture for interpreting their results. To be fit for publication in Nature Comms, the authors need to dig deeper into their results in the following major ways, to provide the field with more insight in the prediction of multiple conformations.

We thank the reviewer for the excellent suggestions made here that we think have greatly improved the manuscript.

The work is also in general lacking connections to the biological function of the discussed proteins, and this needs to be significantly improved in my view to be fit for publication. For instance, the example of perforin is used frequently throughout the paper as an example of predicting multiple conformations. The structure 7PAG is one subunit of a symmetric multimer (see <https://www.science.org/doi/10.1126/sciadv.abk3147>, Fig 1C). If 7PAG was in the test set, then this means that learning from [primarily] monomers enabled Cfold (and AF2) to predict structure in both a monomeric and multimeric context.

We thank the reviewer for this suggestion and have analysed the biological relevance of what induces the conformational changes for the 52 structures that could be predicted with a TM-score>0.8 for both train and test conformations. We have added a section “Biological relevance of sampled conformations” about this.

The biological relevance of sampled conformations

To connect the sampled conformational changes with biological relevance, we analysed the structures that could be predicted with a TM-score>0.8 for both train and test conformations among the samples taken (52 structures, Figure 1c). Most conformational changes are induced by ligand binding (n=42, Supplementary Table 1), a few are due to introduced

mutations (n=5), 3 are due to protein binding and for 2 of them, the trigger is unknown, but may be due to ligands which were not resolved in the X-ray data.

The structural changes induced by ligands (mainly binding of substrates to enzymes or transporters) that are important for cellular functions in one way or the other should have states that are accessible through the analysis of coevolutionary information. However, some conformational variation should not be possible to capture through coevolutionary information alone (e.g. elongation factor Tu which gets its conformational variation from binding different ligands, PDB IDs 2C78:<https://www.rcsb.org/structure/2C78> and 1HA3:<https://www.rcsb.org/structure/1HA3>). This suggests that variation in coevolutionary information may be less important than previously suggested [5,7] in exploring possible alternative conformations and that this process is instead stochastic with different states being constrained by coevolutionary inferred contacts (see below).

We do not find that we can predict conformational changes induced by multimeric interactions with Cfold and note that Perforin is not predicted accurately in both states. The structure the reviewer refers to in Figure 2 is there to illustrate differences between different types of conformational changes but does not refer to predictions. We have underlined this in the figure legend and hope that this makes things more clear: *Different types of conformational changes displayed from native structures*;

None of the above is described in the paper currently.

1 It has been posited by others [see, for instance, introduction of <https://www.nature.com/articles/s41586-023-06832-9>] that dropout in AF2 is able to sample multiple conformations when evolutionary couplings for both structures are present at the level of an entire MSA, and dropout or subsampling an MSA obstructs one set of contacts, allowing for the other to dominate driving the prediction. As long ago as 2012 in Hopf et al. Cell, it was shown that evolutionary couplings were present for multiple sets of contacts of ion channels.

The authors have the potential to evaluate if this is true more generally using their network. For a given protein where both conformations are correctly predicted, can the authors discern if embeddings from the evoformer that lead to either state are significantly different before they are fed into the structure model? If so, this indicates that the evoformer has already learned differentiating representations for both. If not, it would indicate that prediction of both states is stochastic.

We thank the reviewer for this great suggestion. To answer the question we reran the 52 structures that could be predicted with a TM-score>0.8 for both train and test conformations with MSA sampling with the difference of intercepting the embeddings

(single sequence embeddings and pair embeddings) before entering the structure module. We find no relationship with the embedding differences suggesting that the outcome is stochastic - at least in relation to the embeddings that enter the structural module.

We have added a section about this in the main text.

TM-score difference to test conformation vs embedding difference for predictions with the highest TM-scores to the test and train conformations, respectively (n=52). **a)** Comparison between TM-score and cosine similarity of the single sequence embeddings. **b)** Comparison between TM-score and L2 difference of the pair embeddings.

2) Dropout is statistically principled to represent underlying uncertainty within a model, see, for instance, <https://arxiv.org/abs/1506.02142>. Therefore, I was intrigued to see sampling comparing dropout and MSA clustering, which is in effect taking fewer random sequences from across the MSA. These two sampling methods should have fundamentally different theoretical bases, with dropout representing uncertainty across the entire model and MSA clustering adding noise at the sequence level. With this in mind – I want to know more about the proteins that showed a difference between MSA clustering and dropout, and if increased sampling there recovers the failed predictions from dropout. If not, can the authors identify why these worked for MSA clustering and not dropout?

We thank the reviewer for this suggestion. To answer this question we have selected the proteins that could be predicted with a TM-score above 0.8 for both conformations using clustering but not for dropout (n=8). We reran these predictions and increased the number of samples to 1000. We find that none of these can be 'rescued' and conclude that the MSA cluster variation seems superior in introducing stochasticity compared to Dropout.

The only suggestion we have is that the MSA sampling results in more ordered coevolutionary information that better represents different states which evidently is unlikely to obtain by simply activating dropout. According to the embedding analysis, this can't be true though since this shows that the outcome is stochastic. Therefore, we suggest that the MSA sampling rather acts in reverse and results in increased stochasticity increasing the probability of capturing a more diverse set of outcomes (conformations). Note that it is also possible that the network

uses the embeddings in a different way than what is captured by cosine similarity. Analysing how neural networks is not really possible.

We have detailed these findings under “Increasing sampling with dropout to rescue failed predictions” in the supplementary information.

Sampling could also be used to interpret the AF2/Cfold framework. For instance, can the authors use dropout just in some parts of the model to pinpoint more precisely how the AF2 framework is leading to predicting multiple conformations? (i.e., is it in the pair representations? Is it in the first vs 10th evoformer block?) This is probably beyond the scope of this paper, but just floating the idea.

We thank the reviewer for this suggestion and agree that this may be something to investigate in future studies to explore more of the conformational space.

3) The authors could do more in their interpretation of the process of sampling. My guess is that this sampling is far from sampling a Boltzmann distribution from each of the targets, but just how far? Specifically it would be interesting to report what proportion of samples correspond to the trained, vs alternate state, vs. other states. It should be quite possible to compare these to literature experimental values for at least a handful. This would be an opportunity to dig more into biological insight in AF2 + multiple conformation prediction.

We agree with the reviewer that it is unlikely that the evolutionary sampling follows a Boltzmann distribution. We also do not quite know how to compare the samples to literature values directly (or where to find such values that correspond to different conformations). In addition, there is, to our knowledge, no way to test if these samples are truly Boltzmann distributed. What we can say is how likely it is that known conformations are observed from the samples.

To this end, we provide the distribution of TM-scores to the train/test states and can conclude that the test conformations seem to be more likely and that in most cases either the train or test conformation is preferred. As a Boltzmann distribution is an exponential distribution, it is easy to see that the scores do not follow such a distribution as a whole. In individual cases (see below), some samples appear to be approximately exponential towards the test conformations.

We have not included this in the manuscript as we do not think this is relevant for the structure prediction of alternative conformations.

Distribution of conformational samples

TM-score distributions to conformations in the training set and TM-score to unseen conformations (test) using the best strategy (MSA clustering) for structures that could be predicted with a TM-score > 0.8 for both train and test conformations among the samples taken are displayed (52 structures, n=5408 sampled predictions). The test conformations are preferred overall with a median TM-score of 0.86 vs 0.82 for the train conformations.

Distributions. Individual TM-score distributions to conformations in the training set and TM-score to unseen conformations (test) using the best strategy (MSA clustering). Only structures that could be predicted with a TM-score > 0.8 for both train and test conformations among the samples taken are displayed (n=104 samples per distribution).

Distributions. Individual TM-score distributions to conformations in the training set and TM-score to unseen conformations (test) using the best strategy (MSA clustering). Only structures that could be predicted with a TM-score > 0.8 for both train and test conformations among the samples taken are displayed (n=104 samples per distribution).

We have added the fraction of samples that correspond well to either train/test conformations from Figure 1c highlighting that the test conformations are more likely to sample: From all samples, 37% correspond well (TM-score > 0.8) to the test conformations, 33% to the train and 30% to neither of the conformations.

4) Fig 1c presents an interesting result – if I understand right, it seems that some proteins resulted in the “seen” conformation being sampled more, whereas others result in the “unseen” conformation being sampled more. Can the authors interpret any attributes of what causes proteins to fall into either of these extremes? This would be another potential place for interesting insight.

We thank the reviewer for this suggestion. To try and answer the question we selected five examples where the train TM-scores are higher and five where the test TM-scores are. We analysed the conformational types and the biological relevance and found that the length seems to explain the differences the best. The cases where the train conformations are better contain slightly bigger proteins compared to when the test conformations are better (344 vs 318 amino acids). We added a section “Favourable conformations” in the supplementary information.

5) I recognize that Fig. 4 is making the case that pLDDT is not predictive of model accuracy, but the authors need to discuss / show statistics on pLDDT vs tmscore for within the sampling for each member of the test set. If pLDDT is not correlated with TMscore, can the authors provide any insights/takeaways on how to select models for prospective tests?

We thank the reviewer for this suggestion. To further highlight the finding that pLDDT can't be used to select accurate states we display the same plot as in Fig 1c but as a scatter coloured by pLDDT. As can be seen here, the pLDDT score is highly variable and does not seem to select for either conformation overall:

We do not have any suggestion for how to select for either conformation, but we do note that the test conformations seem to be more likely to be predicted. Therefore, a possible strategy for selection is to simply choose the most predicted structure by comparing all structures to each other as can be read from “Distribution of conformational samples”.

6) Depending on how resource intensive training is, the authors could use their setup to answer outstanding questions on how much “regular” AF2 has memorized multiple conformations by comparing the current Cfold to a second re-training of Cfold with all the structure clusters in the train set. This is not a hard ask from me, but if it were easy to re-run it would elevate the impact of the paper.

This is a good idea, but we are not sure that this would answer the question definitively. The reason is that there is still some stochasticity in the predictions due to the sampling of sequences from the MSA which will impact training. Therefore, we suggest that it is not possible to check (definitively) how well structures have been memorised but it is possible to check if alternative conformations that have not been seen can be predicted well (which we do here).

We would like to do this for a future study, but it is currently not easy to retrain Cfold due to migration issues on our cluster resulting in most data having to be reprocessed and due to the computational resources and time necessary for a retraining run making it impossible to meet the review deadline. We appreciate the reviewers' understanding in this matter.

Minor:

Throughout the paper, proteins are referred to by their PDB codes. They need to be referred to by their actual names.

The reason why we use the PDB codes is that this is a requirement from NComms (we have previously done the reverse and then had to change to all PDB codes). We have added the names to Figure 2 displaying the different types of conformational changes as well so that both are now available

I got hung up on when TM-score is referring to TM-score to the ground truth and TM-score between multiple conformations. Would recommend finding a way to better differentiate phrasing.

We thank the reviewer for this suggestion. There is only one instance where the TM-score does not refer to the ground truth in the main text which is when we analyse the structural change between conformations to show that it is harder to predict larger conformational changes. We have stated there in the first sentence that “The structural change (TM-score between conformations)” and do not have any ideas of how to make this more clear. This is also described in the figure legend.

“the structural change between conformations is 0.2-0.3 TM-score (Figure 6)” I don't understand how 0.2-0.3 TMscore is represented in Fig 6. To me it looks like most TM-scores are around 0.8? Please clarify the writing.

We understand that this is confusing and thank the reviewer for this suggestion. It is the TM-score btw conformations → change = 1-TM-score. We have made this more clear in the figure legend:

“As most conformations are between 0.7-0.8, this suggests that the structural change is mainly 0.2-0.3 TM-score.”

The writing is overly simplistic at times:

“Conformational changes are direct effects of environmental changes” -> this is simply not true, discounted by proteins that in vitro interconvert between distinct states. I get the sentiment but it can be made more accurately.

We thank the reviewer for this suggestion and have changed the phrasing to:

“Conformational changes occur due to a variety of reasons, many being direct effects of environmental changes such as interacting with other proteins or substrates.”

“Currently, protein structure is viewed as a single snapshot in time while in reality, it should be viewed rather like a video. This video is the continuous collection of states that make up the protein and its functions.” Again while I agree with the sentiment, this is overly simplistic -- “protein structure is viewed as a single snapshot in time” – 50 years of spectroscopists beg to differ!

We thank the reviewer for this suggestion and have changed the phrasing to:

“Currently, most protein structures are viewed as a single snapshot in time due to only one static conformation being resolved for most structures while in reality, it should be viewed rather like a video.”

Note in methods how long training took and on what sort of hardware.

We have stated the hardware used:

“The effective batch size is 24 distributed across 8 NVIDIA A100 GPUs (3 examples per GPU) with a crop size of 256 residues. “

and updated the methods to include the total training time (training, Methods):

Each step takes approximately 19.7 s, resulting in a total training time of 17 days.

REVIEWER COMMENTS

Reviewer #1 (Remarks to the Author):

The revised version addresses my comments. Thank you.

Reviewer #2 (Remarks to the Author):

Firstly, I apologize for not having realized the following major concern in the first round of review. Continued thought, and reading the additions to the paper after one round of review, have brought it to my attention. It is possible I have missed something very fundamental in the work that already addresses this, if so, I apologize, and rewriting would help regardless to clarify.

If I understand correctly, the paper's main point is that AF2 can predict multiple conformations by virtue of something intrinsic, rather than explicitly having been trained on multiple conformations per protein from the PDB.

30% sequence similarity is a common rule of thumb cutoff for detecting homology, but many homologs can be found that have less than 30% sequence similarity (c.f. <https://www.ncbi.nlm.nih.gov/pmc/articles/PMC3820096/>).

Here's a concrete example of the concern: let's say there are two kinases that are less than 30% sequence similar. Kinases adopt multiple conformations, two of which are the active and inactive conformations. These are very similar between kinases and many structures of both conformations exist for many kinases. As I understand the current training and evaluation scheme, it would have been possible for Cfold to have been trained on an active conformation of one kinase, and an inactive conformation of another kinase, as the conformations for each protein in the training set were randomly selected. This would make Cfold's ability to predict both states for any given kinase less striking than a scenario where, for instance, only kinase active states had been selected. The same holds for open/closed states of GPCRs, transporters – there are many examples of protein families with multiple conformations where the known solved states are highly similar between proteins that can very reasonably be less than 30% sequence similar. The possibility of transferred learning between structures, if not investigated and understood, lessens the impact of the work and still leaves open the question to what degree AF2 memorizes multiple conformations.

Secondly, the authors' responses to the first round of review indicates ignorance of the current understanding and literature in protein conformational landscapes. I illustrate this in a few points below. Without significant rewriting I do not view the paper as fit for publication in Nature Comms.

1. In analyzing differences between the structure pairs correctly predicted, the authors found that many of the structure clusters in the data contained pairs of apo/holo structures, but the authors incorrectly state that "Most conformational changes are induced by ligand binding". The presence of apo/holo structure pairs says nothing about whether a protein occupies both states prior to ligand binding (conformational selection), or if ligand binding causes the conformational change (induced fit), an elementary principle of biochemistry. The last decades have demonstrated that conformational selection is at play in many diverse protein families. This degree of simplicity in the writing concerns me.

2. In my first round of review, I wrote "It should be quite possible to compare [sampled populations] to literature experimental values for at least a handful." In the response to referees, the authors write, "We also do not quite know how to compare the samples to literature values directly (or where to find such values that correspond to different conformations)." This indicates a lack of understanding of the field and without such comparison, I do not view the work as suitable for publication in Nature

Comms. Observables on populations could come from NMR or from single-molecule studies such as FRET, to name a few sources.

Lastly, the authors wrote in their response, "Analysing how neural networks is not really possible" [sic] is out of touch. Interpreting neural networks is part and parcel of modern computational research, and I thought the intent of this paper.

Dear Editors and Reviewers,

We thank you for your insightful comments and suggestions and here provide a point-by-point response to all raised issues and suggestions in bold. We have updated the manuscript accordingly and marked all changes in red throughout the text.

REVIEWER COMMENTS

Reviewer #1 (Remarks to the Author):

The revised version addresses my comments. Thank you.

Reviewer #2 (Remarks to the Author):

Firstly, I apologize for not having realized the following major concern in the first round of review. Continued thought, and reading the additions to the paper after one round of review, have brought it to my attention. It is possible I have missed something very fundamental in the work that already addresses this, if so, I apologize, and rewriting would help regardless to clarify.

If I understand correctly, the paper's main point is that AF2 can predict multiple conformations by virtue of something intrinsic, rather than explicitly having been trained on multiple conformations per protein from the PDB.

The main idea in this work is that the coevolutionary constraints constrain the structure, but also contain descriptions of different states. If slightly different representations are sampled, this should result in different outputs (structures). Here we investigate if this is true by training a network on a structural split of the PDB to obtain an extensive evaluation of all protein structural clusters that differ substantially (>0.2 TM-score).

30% sequence similarity is a common rule of thumb cutoff for detecting homology, but many homologs can be found that have less than 30% sequence similarity (c.f.

<https://www.ncbi.nlm.nih.gov/pmc/articles/PMC3820096/>.

Here's a concrete example of the concern: let's say there are two kinases that are less than 30% sequence similar. Kinases adopt multiple conformations, two of which are the active and inactive conformations. These are very similar between kinases and many structures of both conformations exist for many kinases. As I understand the current training and evaluation scheme, it would have been possible for Cfold to have been trained on an active conformation of one kinase, and an inactive conformation of another kinase, as the conformations for each protein in the training set were randomly selected. This would make Cfold's ability to predict both states for any given kinase less striking than a scenario where, for instance, only kinase active states had been selected. The same holds for open/closed states of GPCRs, transporters – there are many examples of protein families with multiple conformations where the known solved states are highly similar between proteins that can

very reasonably be less than 30% sequence similar. The possibility of transferred learning between structures, if not investigated and understood, lessens the impact of the work and still leaves open the question to what degree AF2 memorizes multiple conformations.

We thank the reviewer for this example. We have tried to clarify what we have done in terms of structural alignment with TM-align and manual checks in the methods section. We have selected structures that have large conformational changes (>0.2 TM-score) to ensure that the structures are indeed different between training and test clusters. As the reviewer mentions, there may be some sequences that have a very low sequence identity but have similar structures and the only possibility to account for these would be pairwise structural alignment of the entire PDB. Since this is too computationally expensive, we have adopted a hybrid strategy where we use 30% sequence clustering to reduce the number of necessary structural alignments. We have performed the structural alignments to the best of our ability and do not know of any more stringent condition or previous study that is certain that there is no structural similarity between any sequences. Although different sequence clusters may have similar structures, it is highly unlikely that these encompass different conformational changes with >0.2 TM-score and that these would happen to be divided in a way that would bias the study given how rare these conformations are (*current pool of 244 conformations out of the 10116 sequence clusters (2.4%)*) among all structures in the PDB.

Secondly, the authors' responses to the first round of review indicates ignorance of the current understanding and literature in protein conformational landscapes. I illustrate this in a few points below. Without significant rewriting I do not view the paper as fit for publication in Nature Comms.

1. In analyzing differences between the structure pairs correctly predicted, the authors found that many of the structure clusters in the data contained pairs of apo/holo structures, but the authors incorrectly state that "Most conformational changes are induced by ligand binding". The presence of apo/holo structure pairs says nothing about whether a protein occupies both states prior to ligand binding (conformational selection), or if ligand binding causes the conformational change (induced fit), an elementary principle of biochemistry. The last decades have demonstrated that conformational selection is at play in many diverse protein families. This degree of simplicity in the writing concerns me.

We thank the reviewer for this comment and have rewritten this section to better reflect the possibilities described by the reviewer:

"Most structures that display conformational changes interact with a ligand (ligand binding, n=42, Supplementary Table 1), a few are due to introduced mutations (n=5), 3 bind to proteins and for 2 of them, the reason for the conformational variability is unknown, but may be due to ligands which were not resolved in the X-ray data."

2. In my first round of review, I wrote "It should be quite possible to compare [sampled populations] to literature experimental values for at least a handful." In the response to referees, the authors write, "We also do not quite know how to compare the samples to literature values directly (or where to find such values that correspond to different

conformations).” This indicates a lack of understanding of the field and without such comparison, I do not view the work as suitable for publication in Nature Comms. Observables on populations could come from NMR or from single-molecule studies such as FRET, to name a few sources.

We have perhaps misinterpreted the reviewer's previous suggestion here. We thought that this suggestion was about MD trajectories or similarities to these. We did not use any NMR data for training and therefore selected three NMR ensembles with structural fluctuations from the PDB to investigate the possibility of sampling similar fluctuations with Cfold.

We ran Cfold with the clustering strategy for PDB ids:

<https://www.rcsb.org/structure/2M6Q>

<https://www.rcsb.org/structure/7ZK0>

<https://www.rcsb.org/structure/2N4A>

We have added a section in the Supplementary information about this: “Comparison with NMR ensembles”. We compare the states by PCA and projecting the structural variation on the first two PCs of the different Cfold samples in comparison with the NMR ensembles. The ensembles here match poorly with the NMR data, which is not surprising – NMR ensembles typically report flexibility within one state and additionally the statistical uncertainty of matching NMR restraints. Thus, to predict NMR ensembles one would need a more physical model of protein dynamics (e.g. MD simulations), plus a statistical model of the NMR statistical uncertainties. Cfold, on the other hand, is aimed at sampling distinct stable or meta-stable conformations, and it has no physical model of local fluctuations within these states. Likely, a combination of Cfold and MD simulation can achieve a more physical description of protein ensembles.

We have also added a statement about this in the discussion:

We analysed three structures that display fluctuations in NMR ensembles to see what dynamical aspects Cfold can capture (Supplementary information). We find that Cfold does not capture dynamical aspects of proteins and therefore likely not of protein conformations. Cfold can predict distinct conformational states of proteins, but no evidence is found to support the transition between these.

NMR ensembles

Cfold predictions

2N4A

7ZK0

Lastly, the authors wrote in their response, "Analysing how neural networks is not really possible" [sic] is out of touch. Interpreting neural networks is part and parcel of modern computational research, and I thought the intent of this paper.

We agree that interpretable machine learning is an active area of research, however, it is not within the scope of our paper. Here, we focus on analyzing the outputs of the Cfold model and compare them with structural biology data.

Reviewers' comments:

Reviewer #2 (Remarks to the Author):

Please see attached pdf for my comments.

We have performed the structural alignments to the best of our ability and do not know of any more stringent condition or previous study that is certain that there is no structural similarity between any sequences. Although different sequence clusters may have similar structures, it is highly unlikely that these encompass different conformational changes with >0.2 TM-score and that these would happen to be divided in a way that would bias the study given how rare these conformations are (current pool of 244 conformations out of the 10116 sequence clusters (2.4%)) among all structures in the PDB.

Thanks to the authors for providing metadata of their train/test splits. (Adding these to the publicly available supporting materials would improve their accessibility.)

From the train / test splits supplied in supplemental information, I was readily able to identify an example of what I was concerned about using the publicly available tool FoldSeek¹ and which the authors have claimed are “highly unlikely.”

One member of the test set is 4Z3N, a MATE transporter from *E. coli*. Using FoldSeek, I readily found a structure, 6IDR, that is highly similar (TM-score = 0.889, would get clustered by the authors’ criteria) with a sequence identity of 21.7% to the structure in the test set.

In test_meta.csv:

1		Entry ID	sequence_cl	structural_cluster
1922	34038	4Z3N	4Z3N	4Z3N
1923	34039	4Z3P	4Z3N	4Z3N
1924	34040	4MLB	4Z3N	4Z3N
1925	34041	21A01	4Z3N	4Z3N

In train_meta.csv:

	Entry ID	sequence_cl	structural_cluster
48715	6IDR	6IDP	6IDP
48716	6IDS	6IDP	6IDP
48717	6IDY	6IDY	6IDY

PDB100 79 hits

Target	Description	Scientific Name	Prob.	Seq. Id.	TM-score
4z3n-assembly1_A	Crystal structure of the MATE transporter C...	Escherichia coli	1.00	100	1.000
4z3p-assembly1_A	MATE transporter ClbM in complex with Rb+	Escherichia coli	1.00	100	0.998
6idr-assembly1_A	Crystal structure of Vibrio cholerae MATE tr...	Vibrio cholerae	1.00	21.7	0.889
3vvo-assembly1_A	Crystal structure of MATE in the bent confo...	Pyrococcus furiosus	1.00	28.1	0.888
6idp-assembly1_A	Crystal structure of Vibrio cholerae MATE tr...	Vibrio cholerae	1.00	21.8	0.887
6ids-assembly1_A	Crystal structure of Vibrio cholerae MATE tr...	Vibrio cholerae	1.00	21.8	0.886
6hfb-assembly3_C	Outward-facing conformation of a multidrug...	Pyrococcus furiosus DSM 3...	1.00	27.2	0.884
3w4t-assembly1_A	Crystal structure of MATE P26A mutant	Pyrococcus furiosus DSM 3...	1.00	27.1	0.884
4mlb-assembly1_A	Reverse polarity of binding pocket suggests...	Pyrococcus furiosus	1.00	27.5	0.884
3vvs-assembly1_A	Crystal structure of MATE in complex with ...	Pyrococcus furiosus	1.00	27.1	0.882
6hfb-assembly1_A	Outward-facing conformation of a multidrug...	Pyrococcus furiosus DSM 3...	1.00	26.9	0.881
3vvo-assembly1_A	Crystal structure of MATE in the straight co...	Pyrococcus furiosus DSM 3...	1.00	27.0	0.880

In test set
In train set

4Z3N: test set

MATE transporter ClbM
from *E. coli*

6IDR: train set

MATE transporter VcmN
from *V. cholera*

TMalign score: 0.889

Sequence similarity: 21.7

Given that doing a principled training split was the main value of the study and the premise upon which the authors' conclusions rest, I no longer view this study to have the value I previously thought it did. The authors would need to do this sort of comparison for each member of their test set to remove any proteins with low-homology, highly-similar structures and revise all their results accordingly.

Regarding NMR ensembles:

The authors have again demonstrated a lack of understanding about biochemistry as well as the current state of the field by incorrectly interpreting multiple conformations in NMR structures as populations. The set of structures contained in a NMR structure model do not reflect populations. See more in ref 2: "The ensemble of models ... does not describe protein dynamics, but rather represents the uncertainty or inconsistencies present in the experimental NMR data."

The authors should be able to identify logical experiments to compare to. They still have not addressed a concern I raised in my first review of comparing sampling to experimental data on populations.

Regarding overall writing

Finally, I still do not view that the authors have substantially improved the writing to reflect a current understanding of the field. I do not intend to enumerate exhaustively here all the instances, but this is one example that has not been substantially changed:

"Currently, most protein structures are viewed as a single snapshot in time due to only one static conformation being resolved for most structures while in reality, it should be viewed rather like a video."

This is still overly simplistic and shows a lack of awareness of what is known in the field.

1. van Kempen M, Kim S, Tumescheit C, Mirdita M, Lee J, Gilchrist CLM, Söding J, and Steinegger M. Fast and accurate protein structure search with Foldseek. Nature Biotechnology, 2023.

2. Vranken, Wim. "NMR structure validation in relation to dynamics and structure determination." Progress in NMR spectroscopy, 2014.
<https://doi.org/10.1016/j.pnmrs.2014.08.001>.

Reviewer #2 (Remarks on code availability):

The supplementary materials were unclear and required further communication to receive materials that allowed me to review it appropriately.

The claims made here regarding the train-test overlap are inaccurate for several reasons outlined below.

- 1. The example provided by the reviewer was never used for testing.**
- 2. The conformations claimed by the reviewer to be identical appear to be functionally different and encompass two distinct structural states of GPCRs. For GPCRs in general, the structural difference of different functional states is less than that of other proteins.**
- 3. No method ever developed (to our knowledge) has ever assessed the pairwise structural similarity across the PDB as this is computationally intractable. Clustering is used on both the sequence and structural level to reduce complexity.**

We outline the response further below in the appropriate sections.

Thanks to the authors for providing metadata of their train/test splits. (Adding these to the publicly available supporting materials would improve their accessibility.)

They are available already through zenodo as stated under data availability. We have also pointed to this link several times during the review process, and we don't understand why the reviewer does not acknowledge that this request is solved:
<https://zenodo.org/records/10837082>

From the train / test splits supplied in supplemental information, I was readily able to identify an example of what I was concerned about using the publicly available tool FoldSeek1 and which the authors have claimed are "highly unlikely."

The test ids used by the reviewer to extract a 'contradictory example' are not the ones we have used for evaluation. Therefore the reviewer's claim that this is a test example demonstrating the failure of FoldSeek1 is not correct. We have stated this in a communication to the editor, where we also attached the test ids used. These are also available in the zenodo repo (see test.tar.zst - test set predictions):
<https://zenodo.org/records/10837082>

One member of the test set is 4Z3N, a MATE transporter from *E. coli*. Using FoldSeek, I readily found a structure, 6IDR, that is highly similar (TM-score = 0.889, would get clustered by the authors' criteria) with a sequence identity of 21.7% to the structure in the test set.

Target	Description	Scientific Name	Prob.	Seq. Id.	TM-score
4z3n-assembly1_A	Crystal structure of the MATE transporter C...	Escherichia coli	1.00	100	1.000
4z3p-assembly1_A	MATE transporter ClbM in complex with Rb+	Escherichia coli	1.00	100	0.998
6idr-assembly1_A	Crystal structure of Vibrio cholerae MATE tr...	Vibrio cholerae	1.00	21.7	0.889
3vvo-assembly1_A	Crystal structure of MATE in the bent confo...	Pyrococcus furiosus	1.00	28.1	0.888
6idp-assembly1_A	Crystal structure of Vibrio cholerae MATE tr...	Vibrio cholerae	1.00	21.8	0.887
6ids-assembly1_A	Crystal structure of Vibrio cholerae MATE tr...	Vibrio cholerae	1.00	21.8	0.886
6hfb-assembly3_C	Outward-facing conformation of a multidrug...	Pyrococcus furiosus DSM 3...	1.00	27.2	0.884
3w4t-assembly1_A	Crystal structure of MATE P26A mutant	Pyrococcus furiosus DSM 3...	1.00	27.1	0.884
4mlb-assembly1_A	Reverse polarity of binding pocket suggests...	Pyrococcus furiosus	1.00	27.5	0.884
3vvs-assembly1_A	Crystal structure of MATE in complex with ...	Pyrococcus furiosus	1.00	27.1	0.882
6hfb-assembly1_A	Outward-facing conformation of a multidrug...	Pyrococcus furiosus DSM 3...	1.00	26.9	0.881

In test set
In train set

4Z3N: test set

MATE transporter ClbM
from *E. coli*

6IDR: train set

MATE transporter VcmN
from *V. cholera*

TMalign score: 0.889
Sequence similarity: 21.7

Given that doing a principled training split was the main value of the study and the premise upon which the authors' conclusions rest, I no longer view this study to have the value I previously thought it did. The authors would need to do this sort of comparison for each member of their test set to remove any proteins with low-homology, highly-similar structures and revise all their results accordingly.

We have performed a 'principled training split' using structural alignment after sequence clustering with following manual inspection as outlined in methods. The example provided by the reviewer was never used for testing (please see the test ids in the final test set that I sent to you or download them from zenodo).

Further, the conformations claimed by the reviewer to be identical appear to be functionally different and encompass two distinct structural states of GPCRs (one is the 'bent form' <https://www.rcsb.org/structure/4z3n> vs <https://www.rcsb.org/structure/6IDR>). For GPCRs in general, the structural difference between functional states is less than that of other proteins.

Regarding NMR ensembles:

The authors have again demonstrated a lack of understanding about biochemistry as well as the current state of the field by incorrectly interpreting multiple conformations in NMR structures as populations. The set of structures contained in a NMR structure model do not reflect populations. See more in ref 2: "The ensemble of models ... does not describe protein dynamics, but rather represents the uncertainty or inconsistencies present in the experimental NMR data."

The authors should be able to identify logical experiments to compare to. They still have not addressed a concern I raised in my first review of comparing sampling to experimental data on populations.

This study was never about addressing structural fluctuations observed in NMR studies, but about predicting distinct conformational states. We do not understand the purpose of the reviewer's determination to include NMR to investigate protein dynamics when the reviewer now suggests that NMR data can't be used for this. We reluctantly agreed to include the NMR data as the reviewer demanded this. That the reviewer now critiques the use of the NMR data to assess dynamical behaviour we find contradictory and would much rather simply remove this data from the study altogether as it was never included in the original submission.

We also find the demeaning comments about our lack of understanding of biochemistry that the reviewer has provided us with throughout the review process concerning. Reviews should be fact-based and constructive.

Finally, I still do not view that the authors have substantially improved the writing to reflect a current understanding of the field. I do not intend to enumerate exhaustively here all the instances, but this is one example that has not been substantially changed:

"Currently, most protein structures are viewed as a single snapshot in time due to only one static conformation being resolved for most structures while in reality, it should be viewed rather like a video."

This is still overly simplistic and shows a lack of awareness of what is known in the field.

We have removed this statement.

REVIEWERS' COMMENTS

Reviewer #1 (Remarks to the Author):

The major issue of concern revolves around possible overlap between training and test sets, in principle a valid concern, but I think the authors clearly responded and dismissed the concerns that were raised. Based on my reading along with the reviewer responses, this appears to be a non-issue.

A second issue is validation and comparison with (experimental) data. That is an important point but fraught with difficulties as there is not a whole lot of data that is immediately useful for the predictions made here. Comparing with NMR ensembles is problematic - as evident from the discussion between the reviewer and authors - and it is clearly well out of scope of this work to carry out new experiments. What is left are suggestions for experimental validation and partial comparison with available conformations in the PDB, both of which were done. There is potentially more that could be done to elaborate in more detail on possible experimental validation and the comparison with examples of alternate states could be extended, but there is probably not too much more that could be done here to address the issue.

The only issue that remains from my perspective is about significance. The main point here is essentially that AlphaFold2 (or its retrained version cfold presented here) can be massaged into producing alternate conformations by varying the MSA input and adding dropout layers. That has been published previously (see the AFSample paper from Wallner). The new point here is a rigorous separation of training and testing with a retrained version of AlphaFold2, but the main conclusion about how to generate conformational states largely remains the same. Moreover, it is not clear from this work that cfold is in fact more useful in the end than simply applying e.g. the AFSample strategy using the original AlphaFold2 model.

Dear Editors and Reviewers,

We thank you for the additional review and your insightful comments and suggestions and here provide a point-by-point response to all raised issues and suggestions in bold. We have updated the manuscript accordingly and marked all changes in red throughout the text.

REVIEWERS' COMMENTS

Reviewer #1 (Remarks to the Author):

The major issue of concern revolves around possible overlap between training and test sets, in principle a valid concern, but I think the authors clearly responded and dismissed the concerns that were raised. Based on my reading along with the reviewer responses, this appears to be a non-issue.

We thank the reviewer for confirming our view.

A second issue is validation and comparison with (experimental) data. That is an important point but fraught with difficulties as there is not a whole lot of data that is immediately useful for the predictions made here. Comparing with NMR ensembles is problematic - as evident from the discussion between the reviewer and authors - and it is clearly well out of scope of this work to carry out new experiments. What is left are suggestions for experimental validation and partial comparison with available conformations in the PDB, both of which were done. There is potentially more that could be done to elaborate in more detail on possible experimental validation and the comparison with examples of alternate states could be extended, but there is probably not too much more that could be done here to address the issue.

We thank the reviewer for this suggestion and have added a paragraph to the discussion about this issue:

Experimental validation of predictions of alternative conformations would provide the ultimate proof of the utility of Cfold. This would entail predicting the structure of possible conformations of proteins where only one conformation is known, selecting for structural variation and subsequent structural determination by e.g. X-ray crystallography. Such a procedure would require substantial resources and is outside the scope of this study. Importantly, alternative conformations may not be accessible unless these are due to a natural structural equilibrium. If conformational differences are induced by cellular environments, these may not be possible to observe unless the exact conditions are also determined making the study of distinct structural states at scale impossible with current technology.

The only issue that remains from my perspective is about significance. The main point here is essentially that AlphaFold2 (or its retrained version cfold presented here) can be massaged into producing alternate conformations by varying the MSA input and adding dropout layers. That has been published previously (see the AFSample paper from Wallner). The new point here is a rigorous separation of training and testing with a retrained version of AlphaFold2, but the main conclusion about how to generate conformational states largely remains the same. Moreover, it is not clear from this work that cfold is in fact more useful in the end than simply applying e.g. the AFSample strategy using the original AlphaFold2 model.

Indeed, Cfold does not present a new way to sample different conformations compared to AlphaFold2. The main novelty here is related to the training regimen which enables us to prove that structure prediction networks such as AlphaFold2 can be used to predict alternative conformational states based on coevolutionary information sampled from an MSA. Previously, no support for this was provided as AlphaFold2 had seen structures identical to or highly similar to all conformations evaluated in e.g. <https://elifesciences.org/articles/75751>.

We thereby provide a foundation for future studies by providing support for the sampling of alternative states being possible and how likely it is to obtain these. This is important as adopting methods that turn out to be false can have a large negative impact on the community. We realise that distinguishing between ML models and how to adequately evaluate them for different tasks is difficult. We do this comprehensively by analysing all known structural classes (n=155 vs n=8 previously) with significant structural changes.

In addition, we analyse the impact of different types of conformational changes on the prediction and showcase what structural changes are predictable (small) and which are not (large). We also show that the prediction of alternative conformations is largely stochastic and can't be directly related to internal network representations explaining how this prediction works.

We have clarified the significance in the results section:

In contrast to previous attempts to predict alternative conformations, we ensure that our structure prediction network (Cfold) does not see any structures similar to those used for evaluation during training. This is the only way to ensure that predicting different conformations is possible and not an artefact from train and test data overlaps.

And added a point in the discussion:

In contrast to previous attempts to predict the structure of different conformations, Cfold is evaluated on a comprehensive set (n=155 vs n=8 previously) of structural states distinct from those seen during training. This is crucial to assess if conformational states can be

predicted and are not simply reproduced from memory (as is the case when using e.g. AlphaFold2).

The significance is also highlighted in the abstract:

...Neural networks such as AlphaFold2 can predict the structure of single-chain proteins with conformations most likely to exist in the PDB. However, almost all protein structures with multiple conformations represented in the PDB have been used while training these models. Therefore, it is unclear whether alternative protein conformations can be genuinely predicted using these networks, or if they are simply reproduced from memory. ...

and in the introduction:

... These protocols are evaluated on very few structures (eight, six and five, respectively) whose sequences may be present in the AF training set. Therefore, it is not known if these alternative conformations are already encoded in the AF embeddings

...
...

To improve over the previous analyses and provide an answer to whether different conformations can be predicted, we first extracted a set of structures from the PDB [14] that have alternative conformations with substantial changes (a difference in TM-score [15] of at least 0.2 between structures) and are not homologous to the training set of AF. This resulted in a total of only 38 proteins with alternative conformations. We do not think this amount is sufficient to address the multi-conformation prediction problem as the total number of structural clusters (TM-score \geq 0.8 within each cluster) is 6696, meaning that only 0.6% of possible structures would be evaluated.

Therefore, we created a dataset suitable for the multi-conformation prediction task by performing a conformational split of the PDB using structural clusters (TM-score \geq 0.8). Thereby, we obtain 244 alternative conformations for evaluation which represent all sequences that have nonredundant structures that differ >0.2 in TM-score in the PDB. As AF (and other structure prediction methods) can't be evaluated on this set due to having seen most of these conformations during training, we train a new version of AF on the conformational split which we name Cfold.

...
...